# Perinatal serotonin signalling dynamically influences the development of cortical GABAergic circuits with consequences for lifelong sensory encoding

Serotonin plays a prominent role in neurodevelopment, regulating processes from cell division to synaptic connectivity. Clinical studies suggest that alterations in serotonin signalling such as genetic polymorphisms or anti-depressant exposure during pregnancy are risk factors for neurodevelopmental disorders. However, an understanding of how dysfunctional neuromodulation alters systems level activity over neocortical development is lacking. Here, we use a longitudinal imaging approach to investigate how genetics, pharmacology, and aversive experience disrupt state-dependent serotonin signalling with pathological consequences for sensory processing. We find that all three factors lead to increased neocortical serotonin levels during the initial postnatal period. Genetic deletion of the serotonin transporter or antidepressant dosing results in a switch from hypo- to hyper-cortical activity that arises as a consequence of altered cortical GABAergic microcircuitry. However, the trajectories of these manipulations differ with postnatal exposure to antidepressants having effects on adult sensory encoding. The latter is not seen in the genetic model despite a similar early phenotype, and a distinct influence of maternal genotype on the development of supragranular layers. These results reveal the dynamics and critical nature of serotonin signalling during perinatal life; pharmacological targeting of which can have profound life-long consequences for cognitive development of the offspring.

Mammalian neurodevelopment is a protracted process, initially dependent on genetic and molecular cues, that then builds on spontaneous and sensory-evoked activity to create a nervous system equipped to generate adaptive behaviour[1–6]. Within this framework, our understanding of the postnatal development of sensory processing within neocortex has primarily been instructed through deprivation and/or lesion studies[6]. These have demonstrated an important role for bottom-up sensory inputs that are—at a later age—integrated with top-down internal predictive influences[7,8]. However, how such

mechanisms function in the context of normal emergent behaviour is poorly understood, but hard to investigate given that sleep is the dominant and likely instructive behavioural state of neonates[9].

We set out to test the hypothesis that neuromodulatory signals that likely fluctuate with the sleep-wake cycle[10] impinge on local circuit elements to structure information transfer in perinatal neocortex. Specifically, we explored the role of serotonin, as while this neuro-modulator occupies an important position at the intersection of behavioural state and sensory experience in adults, it is tightly

✉e-mail: simon.butt@dpag.ox.ac.uk

regulated in neonates[11]. In adults, serotonin signalling oscillates with sleep-wake cycle, with the lowest levels associated with active sleep[10,12,13]–the predominant state in neonates[14]. During adult wake, the environment further dictates serotonin fluctuations through sensation[15], reward[16] and punishment[16]. Further, serotonin signalling represents a major pharmacological target of modern medicine, with selective serotonin reuptake inhibitors (SSRIs) being the most commonly prescribed antidepressants[17].

In early life, the mammalian serotonergic system undergoes significant changes in terms of axonal innervation of cortex[18], neurophysiology of the dorsal raphe nucleus neurons[19], and expression of

the serotonin transporter (SERT). In human[20] and murine[11,21] neurodevelopment, SERT is transiently expressed in a number of distinct populations of non-serotonergic forebrain neurons, including thalamocortical projections to primary sensory areas and a subset of infragranular pyramidal cells in association cortices and hippocampus[11]. In mice, this transient SERT expression emerges during early embryonic development (embryonic day (E)15 in thalamus[11]) but disappears in first-order thalamic nuclei (including the ventral posteromedial nucleus) around P10, prior to the onset of active sensory awareness. Some SERT mRNA expression continues in other regions as late as the second postnatal week[22–25]. This widespread expression

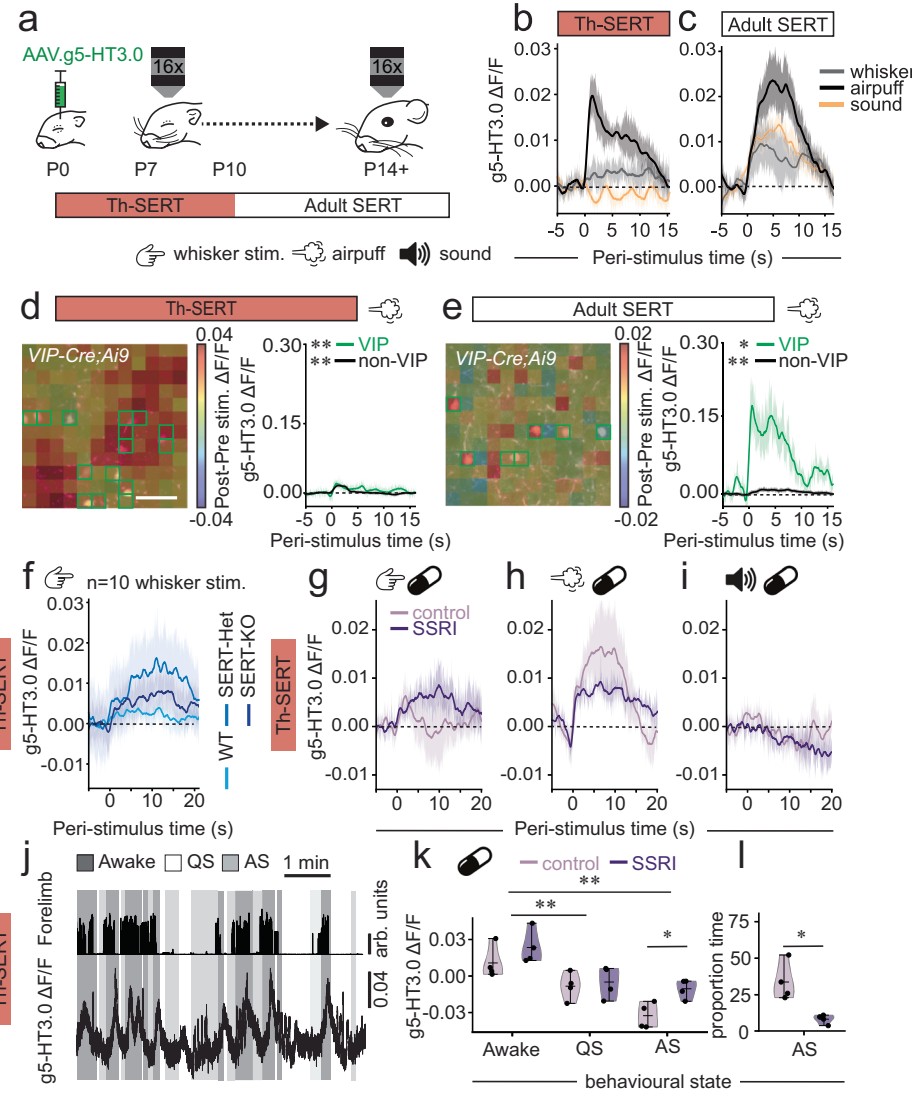

**Fig. 1 | Postnatal serotonin signalling fluctuates with behavioural state and adverse experience, but sensory-evoked responses are clamped by SERT expression. a** Schematic of experimental timeline. **b** g5-HT3.0 signal following whisker ($p = 0.189$), auditory ($p = 0.976$), or airpuff ($p = 0.031$) stimuli during the Th-SERT period ($n = 6$ mice). **c** Corresponding data for the period of Adult SERT expression (P11-13): whisker ($p = 0.046$), auditory ($p = 0.010$), air puff ($p = 0.003$) stimuli ($n = 6$ mice). **d** (Left) change in g5-HT3.0 signal following air puffs during Th-SERT. Right, responses in subregions (green squares) containing ($p = 0.007$) or lacking ($p < 0.001$) VIP interneurons ($n = 38$ VIP cells; 4 mice). **e** Corresponding data for an animal recorded during Adult SERT. Right, response in subregions containing VIP interneurons ($p = 0.043$) or without ($p < 0.001$) ($n = 74$ cells; 4 mice). **f** Peri-stimulus triggered g5-HT3.0 signal in WT ($p > 0.05$, $n = 6$ mice), SERT-Het ($p < 0.05$ for 14 s post-stimulus, $n = 3$ mice) and SERT-KO mice ($p < 0.05$ for 15 s post-stimulus, $n = 4$ mice). **g, h, i** g5-HT3.0 signal of sucrose- ($n = 4$) and SSRI-treated ($n = 4$) Wt

pups following whisker (**g**, control: $p < 0.05$ for 1 s; SSRI: $p < 0.05$ for 3 s), air puff (**h**, control: $p < 0.05$ for 4 s; SSRI: $p < 0.05$ for 20 s) or auditory (**i**, not significant for control/SSRI) stimuli during Th-SERT. **j** Forelimb movement and g5-HT3.0 S1BF signal during wakefulness (Awake; dark grey), quiet sleep (QS; white), and active sleep (AS; light grey). **k** g5-HT3.0 signal during awake, QS and AS for sucrose-control/SSRIs animals (Shapiro−Wilk $p = 0.067$, Two-way ANOVA: sleep state $p < 0.001$, treatment condition $p = 0.017$, interaction state-treatment $p = 0.315$; $n = 4$ sucrose, $n = 4$ SSRI mice). **l** Time spent in AS for control/SSRI-treated mice (Shapiro−Wilk $p = 0.900$, $t$-test $p = 0.015$, $n = 4$ sucrose, 4 SSRI mice). *$p < 0.05$ **$p < 0.01$, paired $t$-test or Wilcoxon signed-rank test of average signal 1 s before vs. after stimulus (except for (**k−l**), following Shapiro−Wilk tests for normality. g5-HT3.0 traces are presented as mean (solid line) ± SEM (shaded area). Violin plots are presented as mean values ± max/min value.

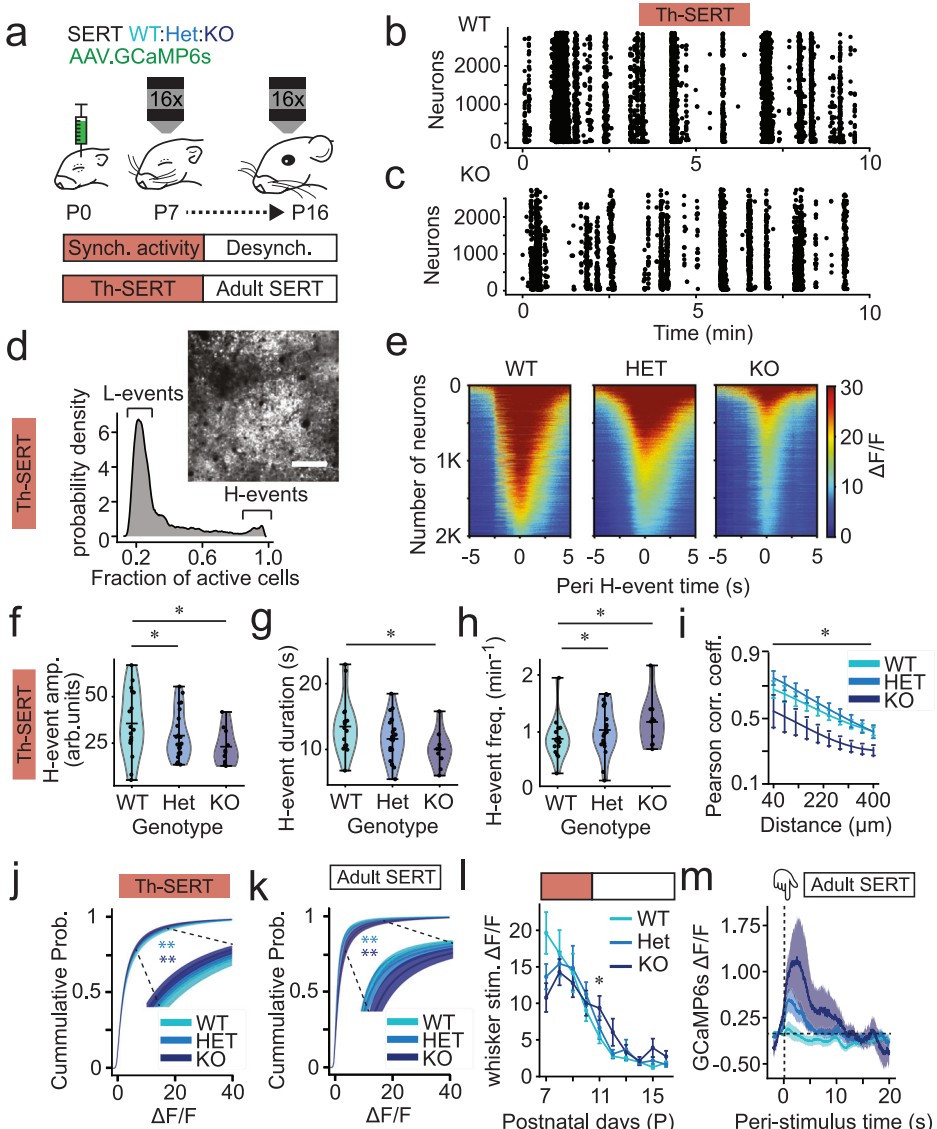

**Fig. 2 | SERT-KO pups transition from hypo- to hyperactivity with the onset of adult-like desynchronous cortical activity. a** Neonates were injected with GCaMP6s in S1BF and imaged P7-16. Raster plot of neuronal spiking across 10 min baseline from (**b**) WT and (**c**) SERT-KO mice recorded at P8 during the Th-SERT period. d, cell co-activation from a single WT animal at P8 demonstrating presence of low (L-) and high (H-) synchronicity calcium events, with inset, a representative H-event; image made from an average of 120 frames (scale bar = 100 μm).
**e** Representative peri (±5 s) H-event GCaMP6s signal heatmaps of 2000 cells in recordings from single WT (left), SERT-Het (centre) and SERT-KO (right) P8 mice; Heat maps represent average signal across all H-events during a 20 min. recording. Mean H-event (**f**) amplitude, (**g**) duration, and (**h**) frequency in WT/SERT-Het/SERT-KO mice during Th-SERT (**f**, Two-way ANOVA genotype p = 0.003. pairwise comparisons: WT-Het p = 0.033; WT-KO p = 0.011; Het-KO p = 0.604; **g**, Two-way ANOVA genotype p = 0.007. Pairwise comparisons: WT-Het p = 0.054; WT-KO p = 0.011; Het-KO p = 0.515. **h** Two-way ANOVA genotype p = 0.001; pairwise comparisons:

WT-Het p = 0.018; WT-KO p = 0.007; Het-KO p = 0.642). **i** Pairwise cell correlations across distance in SERT-WT/Het/KO mice at P8 (p < 0.001). **j, k** Cumulative probability distribution of ΔF/F values in Th-SERT (**j**, Kolmogorov–Smirnov test: WT-HET p = 0.0002; WT-KO p = 0.0007) and Adult SERT (Kolmogorov–Smirnov test: WT-HET p < 0.001; WT-KO p < 0.001). **l** Mean whisker-response amplitude across development in SERT-KO mice showing increases after decorrelation at P11 (Two-way ANOVA genotype p = 0.006, pairwise WT-Het: p = 0.900, WT-KO: p = 0.038, Het-KO: p = 0.0579). **m** Stimulus-triggered average GCaMP6s signal across all cells and mice showing increased whisker-response amplitude at P11-13 (Two-way ANOVA p = 0.040, pairwise WT-HET p = 0.246, WT-KO p = 0.010). Sample sizes for panels (**f–m**) are 19 (WT), 23 (SERT-Het) and 10 (SERT-KO); all other panels (**c–e**) are data from single representative animals. Traces (**m**), error bar (**i, l**) and cumulative probability (j,k) plots are presented as mean ± SEM. Violin plots (**f–h**) are presented as mean values ± max/min value.

suggests tight regulation of serotonin signalling at perinatal ages when bottom-up, sensory-driven instruction of primary sensory areas is ascendent. This makes sense given reports that serotonin attenuates thalamocortical signalling[26–29] while also likely promoting cortico-cortical signalling via ionotropic excitation of 5-HT3aR-expressing GABAergic interneurons[30,31], including the subtype delineated by expression vasoactive intestinal peptide (VIP) that are known mediators of top-down control[32,33].

From a clinical perspective, polymorphisms in the SERT promoter, combined with early trauma, are risk factors for a range of neuropsychiatric disorders[34–38], while SSRIs cross the placenta and *in utero* exposure to SSRIs has been linked to acute serotonin-mediated adverse effects in newborns[39,40]. Moreover, large-scale meta-analyses frequently report that *in utero* exposure to SSRIs is associated with increased risk of neurodevelopmental disorders such as ADHD and autism spectrum disorders (ASD) in human offspring[41]. This

association does not necessarily reflect a causal relationship, with maternal psychiatric diagnosis being one residual confound. To gain additional insights into the necessity of perinatal serotonin signalling, we have used in vivo longitudinal imaging of both serotonin signalling and neuronal activity in both genetic and pharmacological mouse models. We have complemented this with ex vivo slice electrophysiology to probe associated deficits in local neocortical microcircuitry. Combined, these data provide a comprehensive picture and fundamental insights into the role and dynamics of serotonin signalling in the developing brain by identifying critical junctures at which perturbations lead to lifelong alterations in sensory encoding.

## Results

### Postnatal serotonin signalling fluctuates with behavioural state, adverse experience and SERT function

We injected the fluorescence serotonin sensor g5-HT3.0 into the somatosensory cortex (S1BF) of mice in which we labelled a sub-population of 5-HT$_{3a}$R-expressing GABAergic interneuron—those delineated by expression of VIP—with tdTomato (*VIP-Cre;Ai9*). This genetic strategy allowed us to control for optical noise[15] while monitoring serotonin signalling in areas with or without VIP interneurons (Supplementary Fig. S1a). We monitored serotonin signalling dynamics (Supplementary Movie 1) both during the early period of transient expression of SERT in thalamic afferents (Th-SERT; -P10[11,21]) and later, when SERT expression is restricted to raphe afferents[25] (Adult SERT pattern; Fig. 1a and Supplementary Fig. S1b, c). During the Th-SERT period, multi-whisker deflection or presentation of a sound did not elicit a detectable serotonergic response, while an aversive stimulus— an air puff directed to the whisker pad[16] - resulted in a significant time-locked increase in our averaged serotonin signal (Fig. 1b). Following onset of adult-like SERT expression (Adult SERT; P11 to P13), serotonin responses were observed to all stimuli, albeit larger following the aversive airpuff (Fig. 1c); a pattern that continued through juvenile life (Supplementary Fig. S1d). Analysis at the cellular level revealed a change in the spatial distribution of the average serotonin signal to the aversive airpuff from a global response during the Th-SERT (Fig. 1d) to more localised signalling in adult SERT (Fig. 1e), a pattern consistent with developmental changes in serotonergic innervation over this time[18]. It was evident that larger responses were present in areas with VIP interneurons in adult SERT *VIP-Cre;Ai9* mice (Fig. 1e), indicative of targeted release of serotonin at this later age, in line with the reported responses of VIP interneurons to punishment in the adult cortex[42].

These data suggest increased buffering of serotonin during the Th-SERT period, leading to a lack of a detectable response to non-aversive stimuli, but do not rule out underlying developmental changes in 5-HT signalling dynamics. To examine this further, we performed multi-whisker stimulation in SERT knockout mice and observed an increased g5-HT3.0 signal in both heterozygous (SERT-Het) and knockout (SERT-KO) pups but not wild type littermates (Fig. 1f). To examine if this could be replicated with acute postnatal manipulation of SERT function, we dosed pups orally with the serotonin-selective reuptake inhibitor (SSRI) fluoxetine, or the vehicle sucrose, prior to imaging during the Th-SERT period. Under these conditions, multi-whisker stimulation also resulted in a prolonged increase in serotonin signal versus sucrose control (Fig. 1g). Similarly, the aversive stimulus also triggered a longer-lasting response (Fig. 1h), but no change was seen in response to auditory stimulus during this time window (Fig. 1i). Taken together, these results suggest that SERT expression in thalamic afferent likely acts to clamp serotonin release evoked by non-aversive modality-specific stimuli in postnatal S1BF. The lack of early auditory responses is likely explained by the delayed maturation of the auditory system[8], given that detectable responses were observed during the adult SERT period.

This ability to track 5-HT signalling led us to next test whether or not such signalling oscillates with the sleep-wake cycle early in development (Th-SERT period, P8-10), similar to that observed in adults[10,12,13]. Based on previous work relating forelimb movement to particular sleep-wake states[43], sleep was manually scored as a function of movement during infra-red recordings (Fig. 1j, Supplementary Fig. S1e) with continuous movement, immobility, and periods of myoclonic twitches defined as wakefulness, quiet sleep, and active sleep, respectively. Consistent with the adult literature[10,12,13], we observed the highest serotonin signalling levels during wakefulness (Fig. 1j, k). No significant differences were observed in the serotonergic responses to whisker stimulation during different sleep states, albeit there was a trend to an increase during quiet sleep epochs, as sensory stimulation was likely waking the pups up (Supplementary Fig. S1f, g). Next, we tested whether these oscillations were also present in fluoxetine-treated mice and observed higher g5-HT3.0 signal in fluoxetine-treated mice during active sleep (Fig. 1k). This suggests that SERT function is important for the integrity of serotonin fluctuations during the fast bouts of sleep-wake cycles that mice experience at this age. We also report a reduction in active sleep upon SSRI-treatment (Fig. 1l), consistent with a previous study showing that low serotonin is necessary for active sleep[10], which suggests that SSRIs also disrupt sleep architecture. Taken together, these data identify state-dependent regulation of serotonin signalling in postnatal mouse cortex.

### SERT-KO causes early cortical hypoactivity and subsequent hyperexcitability

Next, we explored the consequences of genetically disrupting SERT function on cortical activity in S1BF in vivo using two-photon imaging of GCaMP6s through postnatal development in pups generated by breeding SERT-Het parents (Fig. 2a). During the early postnatal time period (Th-SERT) neocortical activity is characterised by the presence of high- (H-) and low- (L-) synchronicity calcium events (Fig. 2a–e, Supplementary Fig. S2a and Supplementary Movies 2 and 3)[5,6], that then disappear with the onset of desynchronous activity during the second postnatal week[44]. Analysis of these early events (Fig. 2c, d, and Supplementary Fig. S2a) revealed a decrease in the amplitude (Fig. 2f) and duration (Fig. 2g), but an increased occurrence (Fig. 2h) of H-events relative to wild-type (WT), while L-events were unaltered (Supplementary Fig. S2b, d). We also observed a decorrelation of baseline cortical activity across distance in SERT-KO mice (Fig. 2i). The net effect of these alterations was cortical hypoactivity, as evidenced by a left shift of the cumulative probability of ΔF/F values in both SERT-Het and SERT-KO mice (Fig. 2j).

At P11, cortical activity desynchronises (Fig. 2a), concurrent with the disappearance of the transient SERT expression in thalamic afferents and enhanced GABAergic inhibition. Immediately after this time point, we observed a right shift of the cumulative probability of ΔF/F values in both SERT-Het and SERT-KO mice (Fig. 2k), as well as an increase in the amplitude of cortical responses to whisker stimuli (Fig. 2l, m) that persisted through to our final postnatal imaging day (P16) in SERT-KO homozygous mice (Supplementary Fig. S2e–i). This suggests that around P10-P11, cortical dynamics paradoxically transition from hypoactivity to hyperactivity in mutant SERT-KO animals in advance of active sensory awareness.

### GABAergic interneuron activity and underlying synaptic connectivity is altered in SERT-KO mice

The switch from hypo- to hyperactivity in SERT-KO mice occurs at a moment in cortical development, following closure of the L4 critical period of plasticity in S1BF, when bottom-up, thalamic instruction of sensory maps is increasing integrated with cortico-cortical signalling[33] and perisomatic inhibition that decorrelates cortical activity[44,45]. We hypothesised that elevated serotonin levels during the early Th-SERT window could alter this trajectory by attenuating thalamic input— leading to dysregulation of interneurons originating from the

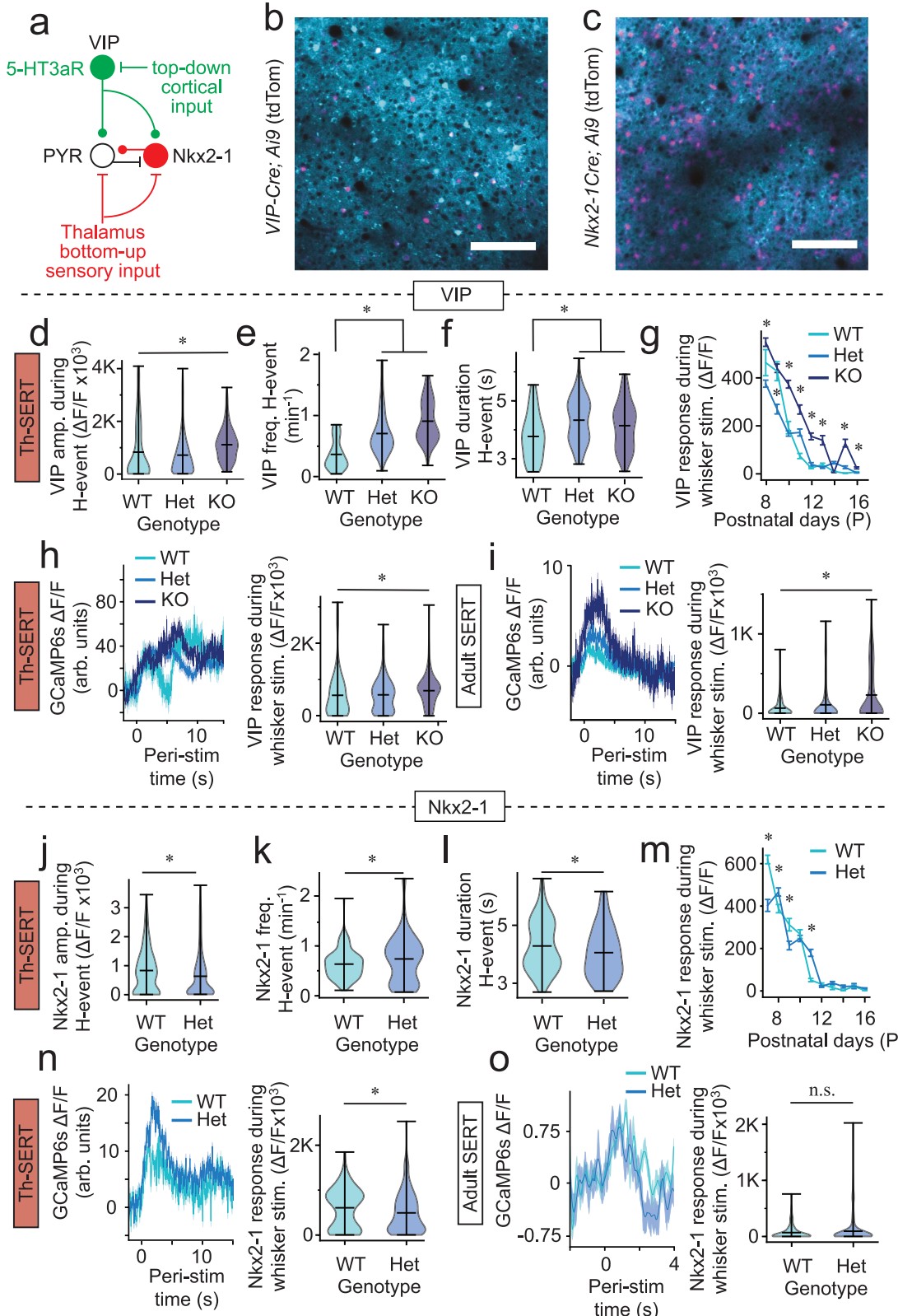

*Nkx2-1*-expressing medial ganglionic eminence (MGE)[46], while at the same time promoting activity in 5-HT3aR-expressing interneurons, including those defined by expression of VIP (Fig. 3a). To test this, we generated pups in which either VIP (Fig. 3b) or Nkx2-1 (Fig. 3c) interneurons were labelled with tdTomato on the SERT-KO background. In vivo imaging revealed that VIP interneurons exhibited higher amplitude calcium transients during H-events (Fig. 3d), as well as elevated

H-event frequency (Fig. 3e), duration (Fig. 3f), and enhanced whisker responses (Fig. 3h, i) in SERT-KO mice compared to WT littermate controls. In contrast, we observed decreased H-event amplitude (Fig. 3j) and duration (Fig. 3l), but increased frequency (Fig. 3k) in Nkx2-1 interneurons in SERT-Het animals. These animals also showed decreased whisker sensory-evoked responses during Th-SERT (Fig. 3m, n), but not Adult SERT (Fig. 3o). No SERT-KO animals born

**Fig. 3 | Altered activity of VIP and Nkx2-1 interneurons on the SERT-KO background. a** Schematic illustrating the contribution of Nkx2-1 and VIP interneuron subtypes to bottom-up, feed-forward and top-down inhibition, respectively. Representative fields of view of mouse S1BF expressing GCaMP6s under the hSyn promoter and tdTomato in (**b**) VIP ($n = 18$ mice imaged) or (**c**) Nkx2-1 ($n = 14$ mice imaged) interneurons (scale bar 100 μm). Violin plots of average (**d**) amplitude, (**e**) frequency, and (**f**) duration of VIP interneuron calcium responses during H-events in SERT-WT/Het/KO mice at P7-10 (**d**, Kruskal−Wallis $p < 0.001$, pairwise WT-HET: $p = 0.087$, WT-KO: $p < 0.001$, HET-KO: $p < 0.001$, **e**, **f**, Kruskal−Wallis $p < 0.001$, pairwise WT-HET: $p < 0.001$, WT-KO: **e**, $p < 0.001$ **f**, $p = 0.037$). **g** Whisker response amplitudes of VIP interneurons in SERT-WT/Het/KO mice from P8 to P16 (*$p < 0.05$ Tukey HSD test). **h, i** VIP whisker response trace (left) and amplitude (right) of SERT-WT/Het/KO mice during the Th-SERT (**h**, Kruskal−Wallis $p < 0.001$, pairwise WT-HET: $p = 0.9612$, WT-KO: $p = 0.029$, HET-KO: $p = 0.004$) and the Adult SERT period (**i**, Kruskal−Wallis $p < 0.047$, pairwise WT-HET: $p = 0.1593$, WT-KO: $p < 0.001$, HET-KO: $p < 0.001$). Violin plot of average (**j**) amplitude, (**k**) frequency, and (**l**) duration of Nkx2-1 interneuron calcium responses during H-events in SERT-WT/Het mice at P7-10 (Mann−Whitney $U$-test **j**, $p < 0.001$; **k**, $p = 0.019$; **l** $p = 0.039$). **m** Whisker response amplitudes of Nkx2-1+ interneurons in SERT-WT/Het mice from P7 to P16 (* $p < 0.05$ Tukey HSD test). Nkx2-1 whisker response trace (left) and amplitude (right) of SERT-WT/Het/KO mice during the Th-SERT (**n**, Mann−Whitney $U$-test $p = 0.001$) and the Adult SERT period (**o**, Mann−Whitney $U$-test $p = 0.053$). VIP sample sizes are 201 cells from 4 animals (WT), 1361 cells from 9 animals (Het), and 546 cells from 5 animals (KO) (**d–i**), while Nkx2-1 are 555 cells from 6 animals (WT) and 1020 cells from 8 animals (Het) (**j–o**). Traces (**h, i, n, o**) and error bar (**g, m**) plots are presented as mean ± SEM. Violin plots are presented as mean values ± max/min value.

on the *Nkx2-1Cre* background survived long enough to allow longitudinal in vivo imaging and analysis.

To better understand the consequences of elevated serotonin on local interneuron circuits, we employed an ex vivo optogenetic approach in S1BF cortical slices (Fig. 4a, b). We conditionally expressed Channelrhodopsin-2 (ChR2) in either VIP or Nkx2-1 interneurons and assessed the (Fig. 4b, e and Supplementary Fig. 3a, b) light (470 nm)-evoked synaptic input onto layer (L)2/3 pyramidal cells (PYRs) across a range of stimulation intensities (LED power); minimal stimulation (50% failure rate) we took as a proxy for unitary monosynaptic input. IPSCs elicited by minimal optical stimulation of VIP interneurons did not vary with genetic background (Fig. 4c and Supplementary Fig. S3a), suggesting no compensation for the increased activity in these cells observed in SERT-KO animals (Fig. 3d–i). In contrast, minimal input onto L2/3 PYRs elicited from Nkx2-1 interneuron was significantly smaller and less variable in SERT-KO versus WT littermates (Fig. 4d and Supplementary Fig. S3b). GABAergic interneurons generated from *Nkx2-1*-expressing ventricular zone represent two main subtypes: parvalbumin (PV)-expressing, multipolar, basket cells and bitufted, somatostatin (SST)-expressing interneurons[47]. To further distinguish between these two subtypes, we used an *SST-Cre* line crossed with the conditional *Ai32(ChR2)* reporter allele, with the thought of using a subtractive approach to resolve the impact on PV interneurons[48] prior to the endogenous expression of this marker in the second postnatal week. These experiments revealed an increase in the minimally-evoked conductance of SST interneuron-evoked IPSCs between WT and SERT-KO L2/3 PYRs (Fig. 4e and Supplementary Fig. S3c). Detailed assessment of the parameters used for optogenetic stimulation of Nkx2-1 versus SST interneurons supports the idea that minimal stimulation of the former elicits largely PV-mediated IPSCs onto pyramidal cells (Supplementary Fig. S3d–j). Therefore, it would appear that elevated serotonin leads to attenuated PV interneuron input onto L2/3 PYRs, most likely through a reduction in thalamic input[26]. To test if this results from a chronic reduction in thalamic afferent input in SERT-KO animals, we used an *Olig3-Cre* driver line[49] to express ChR2 in these fibres (Fig. 4f) and then tested both monosynaptic thalamic EPSCs (Fig. 4g, h) and feed-forward inhibition (Fig. 4i, j) in L4 spiny stellate neurons across the various genotypes. While we found no difference in the strength of unitary thalamic EPSC input between the three backgrounds (Fig. 4h), disynaptic feed-forward inhibition was reduced in SERT-KO pups compared to WT littermates during the Th-SERT period (Fig. 4j)−consistent with the idea that activity-dependent maturation of PV interneurons is compromised in the SERT-KO animals.

Previously, we have identified a transient L5b SST interneuron circuit[50,51] that predates the onset of PV-mediated, fast, feed-forward inhibition (Fig. 4k). To explore if this is also altered in the presence of increased serotonin, we used laser scanning photostimulation (LSPS) of caged glutamate in ex vivo slices to map columnar GABAergic input (Fig. 4l). Intriguingly, in WT pups GABAergic input was almost exclusively local to L4 during the early time period (Fig. 4m) when we would

ordinarily expect the L5b-L4 connection to predominate[50]. In contrast, GABAergic input was almost entirely translaminar in SERT-KO animals (Fig. 4l, m). To such an extent that it would appear that local, intralaminar PV-mediated input is largely absent in these animals, matching our optogenetic assessment (Fig. 4d). Overall, these ex vivo data support a model wherein elevated serotonin in SERT-KO pups negatively impacts the activity-dependent development of PV interneurons during the hypo-active Th-SERT time window. This happens to a point where there is insufficient feed-forward inhibition around the transition to adult desynchronous in vivo activity, resulting in subsequent hyperactivity. The only residual anomaly was the LSPS data from WT pups, which was at odds with our current understanding of transient circuits[50], having previously only been observed following either genetic silencing of SST interneurons or enhancement of PV basket cell synaptic integration. Moreover, we found no difference in the in vivo spontaneous and sensory-evoked activity in WT pups generated from Het mothers versus wild-type animals during the postnatal Th-SERT period (Fig. 4n–q).

## Serotonin influences the developmental trajectory of transient GABAergic circuits in postnatal S1BF

Previous studies have suggested that maternal SERT-Het genotype can lead to altered offspring development and behaviour[52–55]. We reasoned that this might explain the absence of the transient L5b-L4 circuit[50] in WT pups and tested this by crossing SERT-Het males with wild-type females to generate a mix of heterozygous and wild-type pups, termed wt and het (Fig. 5a). LSPS mapping of the offspring revealed the presence of the L5b-L4 connection (Fig. 5b) in both wt and het offspring from this breeding paradigm with a similar profile (Fig. 5c) and developmental trajectory (Fig. 5d) to that previously reported[50]. Comparison of the L5b input between the wt pups ($n = 7$) born from this breeding paradigm versus WT pups generated using our previous SERT-Het x SERT-Het breeding (Fig. 4l) revealed a significant difference in L5b GABAergic input onto L4 spiny stellate neurons (SSNs) ($t$-test, $p = 0.0367$). We further explored total GABAergic input onto supragranular L2/3 neurons in the SERT-Het male and wild-type female cross and identified a novel L5b connection in wt pups at earlier ages (Fig. 5e, f) than previously reported[51]. However, this was absent in the het offspring from the same breeding paradigm, with a difference observed between genotypes until after the end of the first postnatal week (Fig. 5g). Therefore, it would appear that not only does a SERT-Het maternal background alters the trajectory of the early transient interneuron circuit in S1BF (Fig. 4)−most likely via a reduction in the embryo's exposure to serotonin *in utero* given the role of maternal SERT in foetal supply of serotonin through the placenta[56], but postnatal alterations in serotonin signalling in het pups also have consequences for transient, translaminar connectivity mediated by SST interneurons[50,51]. To understand if these changes could be attributed to altered thalamic innervation/signalling or−in the case of the L5b-L4 loop, direct influence of elevated serotonin *in utero*, we used a further

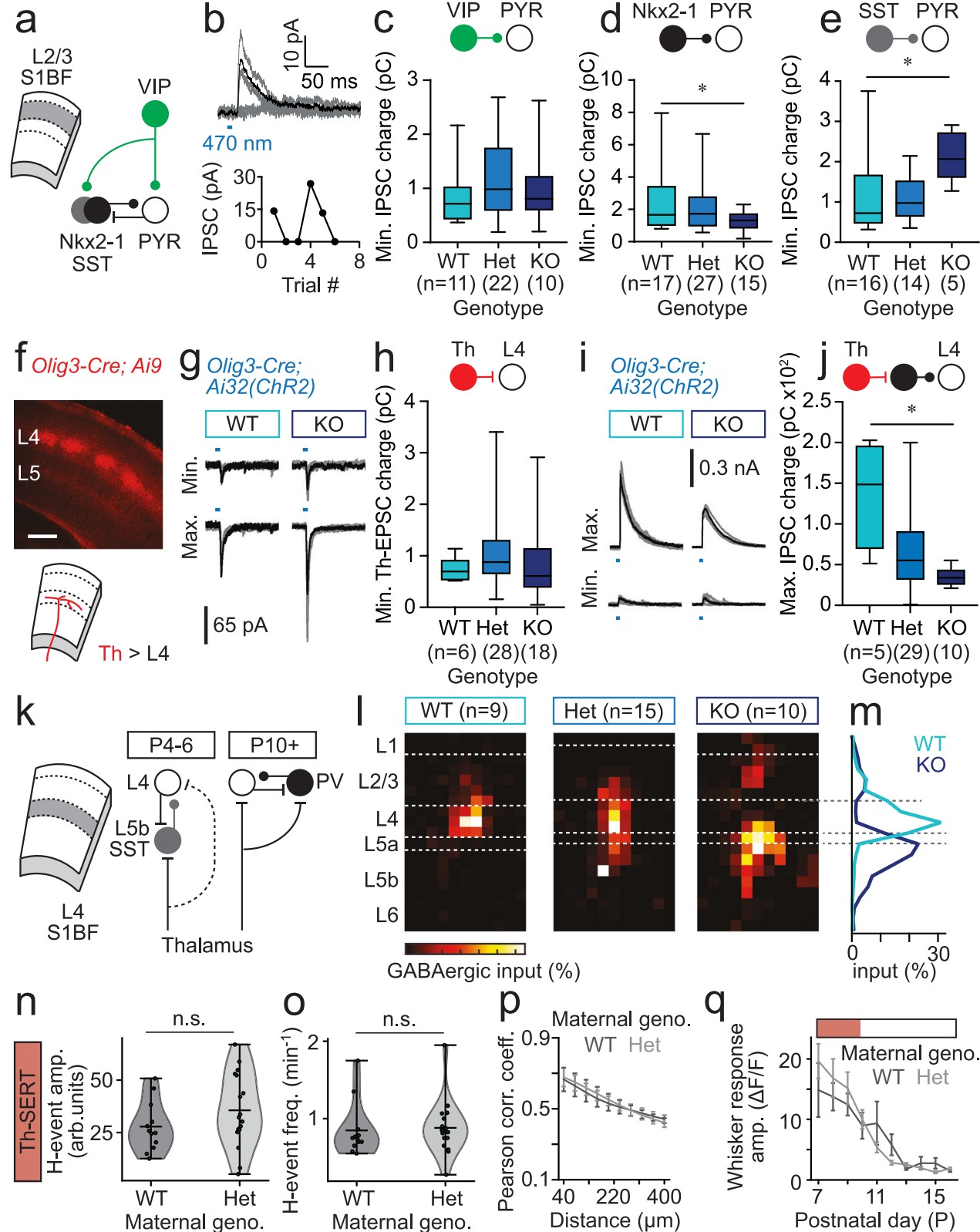

genetic model, *Dlx5/6Cre;Celsr3^fl/fl*, in which *Celsr3*-KO mice lack thalamocortical and corticothalamic connections[57] (Supplementary Fig. S4a, b). LSPS recording in control (Supplementary Fig. S4c) and *Celsr3*-KO littermates revealed a normal developmental trajectory for the transient L5b-L4 connection in the latter (Supplementary Fig. S4d), providing further support that serotonin signalling dynamics across perinatal timepoints is in itself a critical determinant of early

GABAergic circuits in cortex, influencing the formation of translaminar SST connections.

## Postnatal SSRI exposure mirrors genetic disruption of SERT function

Our results point to a critical role for perinatal serotonin in the formation of cortical interneuron microcircuitry. To explore postnatal

**Fig. 4 | Ex vivo characterisation of the impact of SERT deletion on postnatal GABAergic circuits in S1BF. a** schematic of interneuron-pyramidal cell (PYR) circuit in L2/3 of S1BF. **b** Representative minimal optogenetic stimulation data showing interneuron synaptic input onto L2/3 PYRs. Minimal interneuron input onto L2/3 PYRs for **c**, VIP (one-way ANOVA ($F_{(2, 40)} = 0.7044; p = 0.5004$); **d**, Nkx2-1 (one-way ANOVA ($F_{(2, 56)} = 2.781; p = 0.071$. Pairwise comparisons: WT-Het $p = 0.666$; *WT-KO $p = 0.047$); **e**, SST (one-way ANOVA ($F_{(2, 32)} = 4.492$; genotype $p = 0.0191$; Pairwise comparisons: WT-Het p = 0.9733; *WT-KO $p = 0.019$) interneurons. **f** Conditional tdTomato expression in thalamic afferents innervating S1BF (P7 SERT-Het pup)(n = 29 *Olig3-Cre* animals used in experiments); scale bar: 250 μm. **g** Representative minimum (Min.)/maximum (Max.) optogenetically-evoked thalamocortical (Th-)EPSCs recorded in L4 spiny stellate neurons (SSNs) in WT/SERT-KO pups. **h** Minimum Th-EPSC recorded in SSNs across the three genotypes (one-way ANOVA ($F_{(2, 48)} = 0.6741; p = 0.514$). **i** Min./Max. optogenetically-evoked thalamic feed-forward IPSCs recorded at $E_{Glut}$ in SSNs in WT/SERT-KO pups. **j** Max. feed-forward IPSC recorded in SSNs across the three genotypes (one-way

ANOVA ($F_{(2, 41)} = 10.81; p < 0.001$). **k** Schematic of the transient SST input onto SSNs present prior to the onset of PV-mediated feed-forward inhibition. **l** Total GABAergic input onto SSNs across genotypes during the Th-SERT (P4-6) time window. Dashed white lines, average layer boundaries. **m** Average profile of GABAergic input onto SSNs recorded in WT and SERT-KO pups. **n** H-event frequency in WT pups from SERT-WT (n = 12 pups)/Het (n = 17 pups) dams at P7-10. (Mann–Whitney $U$-test $p = 0.277$). **o** GCaMP6s signal amplitude of cells during H-events in WT mice from SERT-WT (n = 12 pups) or SERT-Het (n = 17 pups) dams at P7-10 ($t$-test $p = 0.916$). **p** Pairwise cell correlations across distance in the same pups at P8 (two-way ANOVA, genotype $p = 0.790$, SERT-WT dam: n = 12 pups; SERT-Het dam: n = 17 pups). **q** GCaMP6s whisker response across development (two-way ANOVA, genotype $p = 0.746$, SERT-WT dam: n = 12 pups; SERT-Het dam: n = 17 pups). Box plot whiskers represent minimum to maximum, boxes extend 25th to 75th percentile, centre line is median. Errorbar (**p**, **q**) plots are presented as mean ± SEM. Violin plots (**n**, **o**) are presented as mean values ± max/min value.

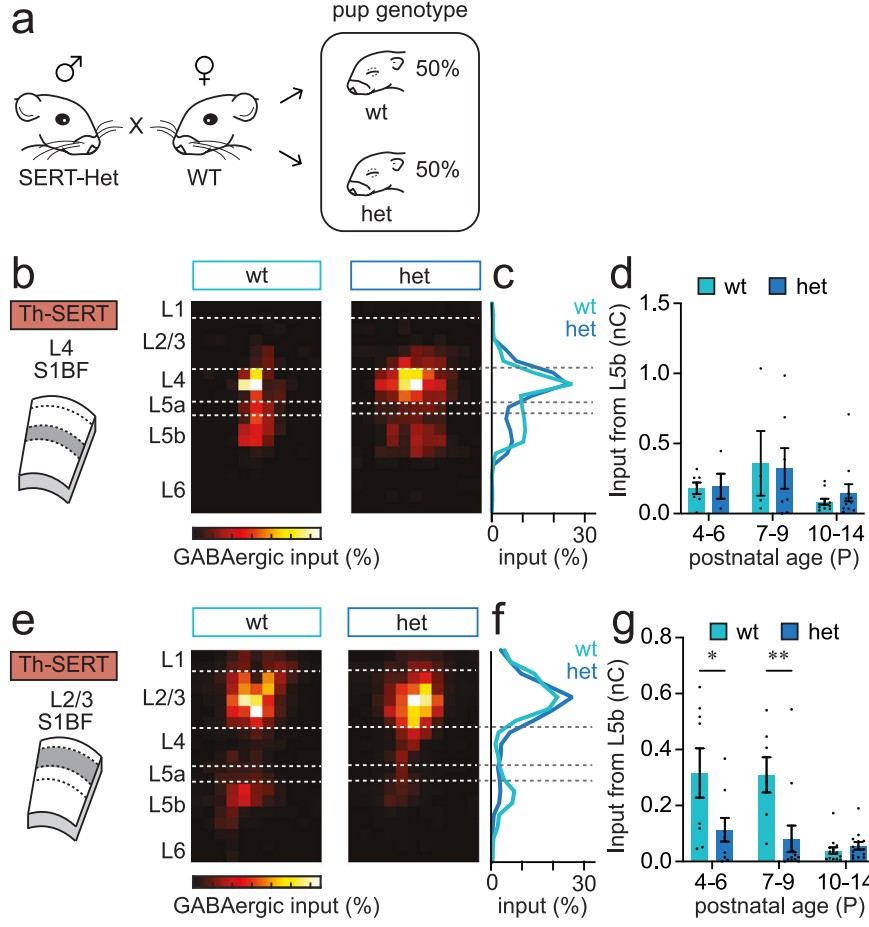

**Fig. 5 | Altered serotonin signalling variously impacts transient, translaminar GABAergic circuits in postnatal S1BF. a** Diagram showing the breeding paradigm used to generate wildtype (wt) or SERT-het (het) pups from wildtype dams. **b** Average LSPS maps for total GABAergic input onto L4 SSNs recorded in pups generated from the breeding paradigm shown in (**a**) mice during the P4-6 time window (wt pups, n = 7 cells; het pups, n = 5 cells). **c** Profile of GABAergic input across the depth of cortex from the maps shown in (**b**). L5b input was present in both WT and het pups born from WT dams. **d** Total L5b GABAergic input onto L4 SSNs recorded in animals bred from SERT-Het male and WT female pairs across

postnatal development, broken down according to the genotype of the pup (wt/het)(mean ± SEM); wt: P4-6, n = 7 cells; P7-9, n = 4; P10-14, n = 11; het: P4-6, n = 4; P7-9, n = 7; P10-14, n = 11. **e** GABAergic input maps for L2/3 PYRs during the P7-9 time window (wt pups, n = 7 cells; het pups, n = 12 cells). **f** Profile of GABAergic input across the depth of cortex from the maps shown in (**e**). **g** timeline of L5b GABAergic innervation of L2/3 PYRs (mean ± SEM); (Two tailed $t$-test: P4-6, $p = 0.048$; P7-9, $p = 0.009$, P10-14, $p = 0.344$); wt: P4-6, n = 8 cells; P7-9, n = 7; P10-14, n = 14; het: P4-6, n = 9; P7-9, n = 12; P10-14, n = 13.

effects, we next switched to dosing of WT mice with the serotonin selective reuptake inhibitor (SSRI) fluoxetine (10 mg/kg of body weight) from P2 to P14 (Fig. 6a); a window previously identified as SSRI-sensitive for cortical development[58,59]. We used the serotonin sensor g5-HT3.0 to confirm that this oral dosing was increasing S1BF

serotonin. We observed increases during both Th-SERT (P7-10) and the immediate Adult SERT period (P11-14), with the latter being of higher magnitude (Supplementary Fig. S5a). This further confirmed the buffering role of the developmental overexpression of SERT in non-5-HT neurons. Then, we explored the effects of SSRI in cortical population

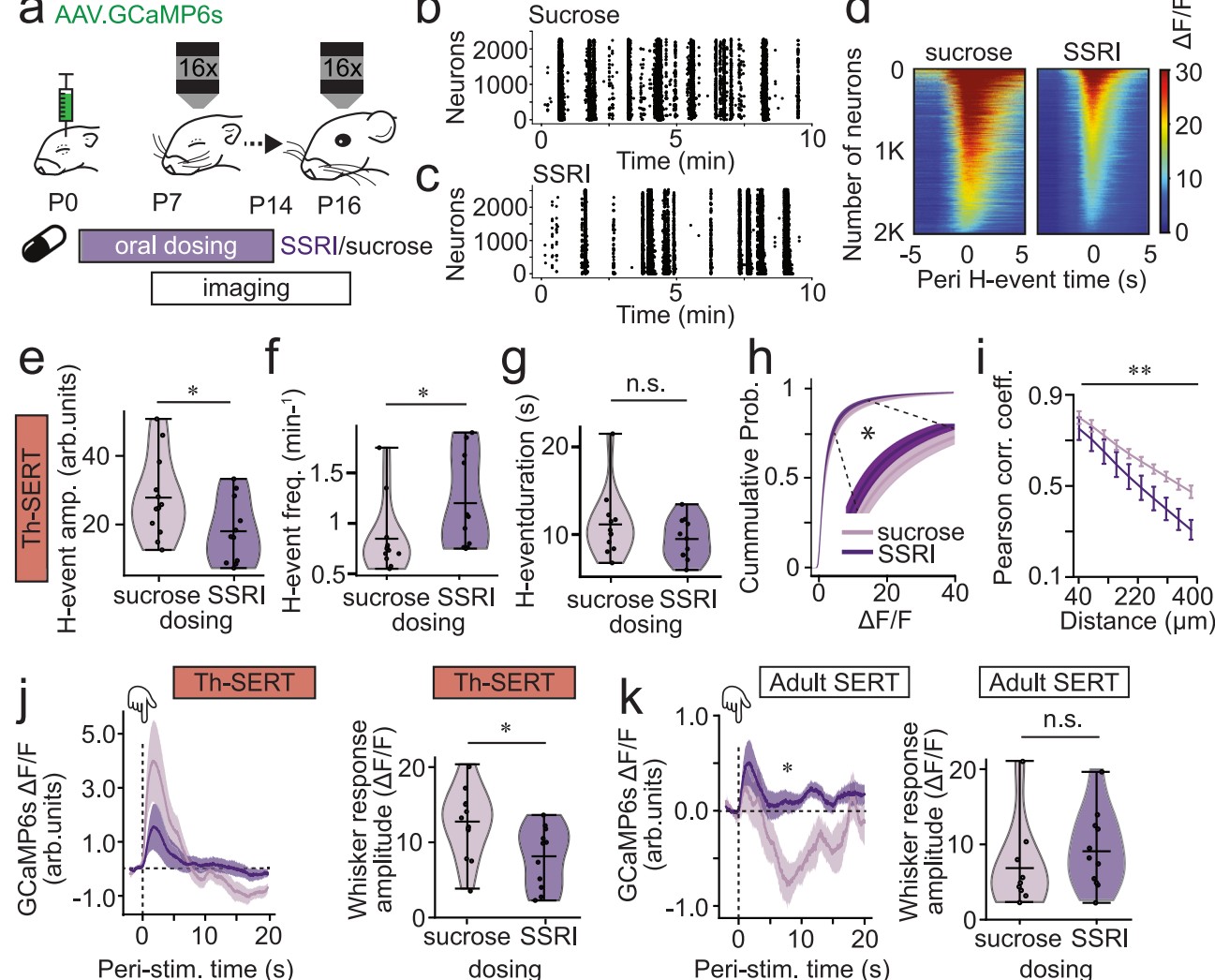

**Fig. 6 | Postnatal fluoxetine treatment recreates the SERT-KO transition from cortical hypoactivity to hyperexcitability. a** Animals were dosed orally with either 10% sucrose (control vehicle) or 10 mg/kg of fluoxetine (SSRI) daily from P2 to P14, injected neonatally with GCaMP6s, implanted with a cranial window and head-fixing plate at P6 and imaged from p7 to P16. Raster plot of neuronal spiking across 10 min baseline from sucrose (**b**) and SSRI (**c**) postnatally-treated representative mice during Th-SERT (P8), illustrating the presence of H- and L-events. **d** Representative peri (±5 s) H-event GCCaMP6s signal heatmaps of 2000 cells in recordings from a single sucrose- (left) and SSRI- (right) treated P8 mouse. Heat map constructed from averaging the signal across all H-events detected on each mouse during a 20 min baseline recording. Violin plot of mean H-event (**e**) amplitude, (**f**) frequency and (**g**) duration in SSRI- versus sucrose-treated mice during Th-

SERT (**e**, *t*-test *p* = 0.040, **f**, Mann–Whitney *U*-test: *p* = 0.016, **g**, Mann–Whitney *U*-test: *p* = 0.310). **h** Cumulative probability of Δ*F/F* 20 min baseline values of all Sucrose/SSRI treated animals at P7-10 (**a**, Kolmogorov–Smirnov test: *p* = 0.022) **i**, Baseline pairwise cell correlations across distance in SSRI-treated mice compared to sucrose-treated littermate controls at P9 (Two-way ANOVA, treatment *p* < 0.001). Stimulus-triggered average trace (left) and response amplitude (right) of GCaMP6s signal across all cells and mice during Th-SERT (**j** *t*-test *p* = 0.018) and Adult SERT (**k**, Mann–Whitney *U*-test, *p* = 0.254), in SSRI- or Sucrose-treated mice. Sample sizes are 12 (Control) and 12 (SSRI). Traces, errorbar and cumulative probability plots are presented as mean ± SEM. Violin plots are presented as mean values ± max/min value.

dynamics using the calcium indicator GCaMP6s. SSRI-treated mice exhibited altered H-events (Fig. 6b–d): a reduction in amplitude (Fig. 6e), an increase in frequency (Fig. 6f), but no change in event duration (Fig. 6g) in SSRI-dosed pups compared to vehicle control (sucrose)-treated littermates. No differences were found in L-events (Supplementary Fig. S5b–d). The net effect of these alterations was early cortical hypoactivity as highlighted by the left shift of the cumulative probability distribution of Δ*F/F* values in SSRI-treated mice (Fig. 6h). We also observed a decreased correlation across distance (Fig. 6i) and a reduction in the amplitude of whisker-stimulus evoked responses during Th-SERT (Fig. 6j). The sum of these observations indicates that both genetic (Fig. 2) and pharmacological (Fig. 6) disruption of SERT cause cortical hypoactivity during the Th-SERT period.

Following the onset of desynchronous cortical activity (~P11), whisker-evoked responses had an increased mean calcium signal but no changes in response amplitude (Fig. 6k) in SSRI- versus sucrose-treated mice. These results suggest cortical hyperexcitability in SSRI-treated mice during the immediate Adult SERT period, albeit to a lesser degree than SERT-KO mice. Of note, sucrose-treated mice exhibited a biphasic sensory response at these ages, with a drop in the calcium signals following the initial sensory response (Fig. 6k). This observation is similar to other reports that have identified increased perisomatic inhibition at this age[45,60].

Analysis of interneuron recruitment in SSRI-treated animals revealed that VIP interneuron have H-events with increased frequency but unaltered amplitude or duration (Supplementary Fig. S5e, g), as well as a decrease in Th-SERT (Supplementary Fig. S5i) response to

whisker stimulation, but an increase during Adult SERT (Supplementary Fig. S5j). Nkx2-1 interneurons showed H-events with increased amplitude and frequency but unchanged duration in SSRI postnatally-treated mice, compared to sucrose-treated littermates (Supplementary Fig. S5k–m). Nkx2-1 interneurons also showed an increased response during Th-SERT whisker response (Supplementary Fig. S5n, o), which based on previous studies, might contribute to the hypoactivity[4]. Interestingly, the impact of altered serotonin on early activity during Th-SERT time window did vary in some regards (e.g., VIP/Nkx2-1 H-event amplitude) between SSRI-treated and SERT-KO mice. Observations that likely stem from the temporal difference of these SERT disruptions (postnatal versus lifelong) and the nature of the disruption (genetic versus pharmacological).

## Postnatal SSRI-dosing results in long-lasting effects on sensory encoding

Altered serotonin signalling during development has been linked to lifelong deficits in sensory processing[37,61]. To explore if this is the case in our models, we studied the effects of developmental disruption of serotonin signalling in adult sensory encoding. We first mapped single barrels using online stimulus-triggered averaging (Supplementary Fig. S6a) and probed adult somatosensory cortical encoding within that field of view by presenting an array of different stimuli: single-whisker, multi-whisker, airpuff, sound, and smooth and rough vibrating surfaces, all of which triggered a sensory response in cortex, except sound (Supplementary Fig. S6b). Single-whisker stimulation revealed increases in cortical response amplitude and neuronal recruitment in SSRI-treated, but not SERT-KO animals (Fig. 7a–e). We observed a similar, but more pronounced, hyper-responsivity upon the presentation of an airpuff (Fig. 7f–i), which was also underlined by an increase in neuronal correlation during airpuff responses (Supplementary Fig. S7). These results are consistent with a prolonged deficit in inhibitory control of information transfer.

To test this hypothesis, we quantified interneuron numbers in S1BF using histology for VIP(tdTomato), Nkx2-1(tdTomato), or PV alongside automated DAPI cell detection (Fig. 7j and Supplementary Fig. S8). Due to the failure of Nkx2-1(tdTomato) SERT-KO animals to survive to adulthood, we were reliant on PV immunohistochemistry (Fig. 7j, k) that identified a decrease in PV expression across the depth of cortex, while the distribution and number of VIP interneurons remained unchanged, in contrast to previous reports[62]. Intriguingly, while SSRI-treated animals showed a parallel decrease in Nkx2-1(tdTomato) interneurons, PV interneuron numbers were not altered. There was also an increase in VIP cells (Fig. 7l) as previously reported[31] in SSRI postnatally-treated mice. This points to a divergence in the outcome of the two models, with acute postnatal SERT disruption having a more pronounced impact on VIP interneuron maturation and survival. The decrease in PV in SERT-KO mice and Nkx2-1 interneurons in SSRI postnatally-treated mice was coherent with both our observed early hypoactivity (Figs. 2 and 6) and previous reports finding that survival of these interneurons is activity-dependent during this period of neurodevelopment[3].

The sum of these data identified developmental serotonin as a critical determinant of population activity in S1BF. To understand how this plays out across the full repertoire of sensory encoding in this cortical area we next analysed the response to an array of stimuli and performed logistic regression (Fig. 7m). Training a classifier on the post-stimulus traces of 500 randomly selected cells revealed that the different types of stimuli could be successfully distinguished from their cortical responses (Supplementary Fig. S6c) with very high accuracy and without significant differences among genotypes or treatment Supplementary Fig. S6d). These results suggest that the discriminability of cortical representation of different stimuli remained intact in both SERT-KO and SSRI-treated mice. We next tested whether animal conditions (genotype/treatment) could be classified based on the responses to the various stimuli; if so, then the sensory representation would be different. The classifier trained on SERT genetic backgrounds did not perform above significance level (Fig. 7n), as defined by bootstrapping statistical analysis (Supplementary Fig. S6f). In contrast, maternal genotype (WT/SERT-Het) could be decoded from adult baseline or sound trials but not whisker-stimulation trials (Fig. 7n). This suggests a possible underlying difference in resting-state dynamics arising from the developmental effects of maternal SERT disruption (i.e., SERT-Het dams), effects that are not observed in population activity in the immediate postnatal period (Fig. 3). Finally, postnatal SSRI treatment was classifiable from all but the multi-whisker stimulation trials, and peaked with airpuff stimulation (Fig. 7n). The latter was consistent with principal component analysis that revealed that treatment-condition had a high separation along their first component (Supplementary Fig. S6e). These data suggest that sensory representation per se is not altered by lifelong genetic disruption of SERT, although it is evident that maternal background can have an influence on baseline—equivalent to resting state—dynamics. In contrast, transient pharmacological disruption during a critical phase of interneuron maturation has prolonged consequences for representation of touch.

## Discussion

Our results show that serotonin signalling is tightly regulated in the perinatal cerebral cortex to ensure appropriate instruction of bottom-up sensory circuits. We find that while serotonin fluctuates with behavioural state in the postnatal cortex in a manner similar to adults, transient SERT expression otherwise ensures sensory serotonergic unresponsiveness with the exception of aversive stimulation (Fig. 1). Disruptions of this early buffering results in decreased cortical activity and altered development of interneuron subtypes involved in bottom-up, feedforward inhibition (Nkx2-1)[50] and top-down inhibitory control (VIP)[32] (Figs. 2–4). As such, serotonin is a critical modulator of activity within cortical microcircuits that reflects both environment and behavioural state, in a manner consistent with the reported role of serotonin in interneuron development[30,31,62,63] and adult sensory cortices[64–66]. Further, it can provide a state-dependent mechanism for the regulation of cortical activity[9]. Disrupted early regulation results in a hyperexcitable sensory cortex that persists into adulthood in mice treated postnatally with an SSRI (Fig. 7). We speculate that this does not happen in SERT-KO mice due to continued alterations in serotonin signalling dynamics. These results provide a longitudinal neurophysiological account that builds on reported alterations in sensory cortices of SERT-KO rodents[28,63,67]. We further observed an effect arising from maternal phenotype[54,68], namely that wildtype mice generated by breeding SERT-het dams exhibited a distinct and subtle change to early supragranular GABAergic innervation[50], one that leads to life-long changes in resting state dynamics (Fig. 7).

Our study takes advantage of recently developed neurotransmitter sensors to probe early serotonin dynamics[12,13]. Deployment of these genetic tools in a developmental setting has the potential to revolutionise our understanding of early cortical dynamics. In our hands, viral delivery of g5-HT3.0 gave sufficient expression by the onset of our imaging time window (P7) to observe large scale serotonin responses to aversive stimuli, as well as lower amplitude time locked average responses to whisker stimulation in pups with altered serotonin signalling—either via genetic deletion of SERT or via SSRI dosing. The latter manipulations likely preclude low levels of expression being the primary driver of the reduced serotonin signal during the Th-SERT period and rather suggest that transient expression of the transporter in thalamic afferent fibres acts to buffer serotonin during a life stage dominated by active sleep[9], when programming of primary sensory cortices is highly dependent on peripheral activity[1].

Our results identify an effect of serotonin in modulating discontinuous activity in vivo in layer 2/3 of postnatal somatosensory

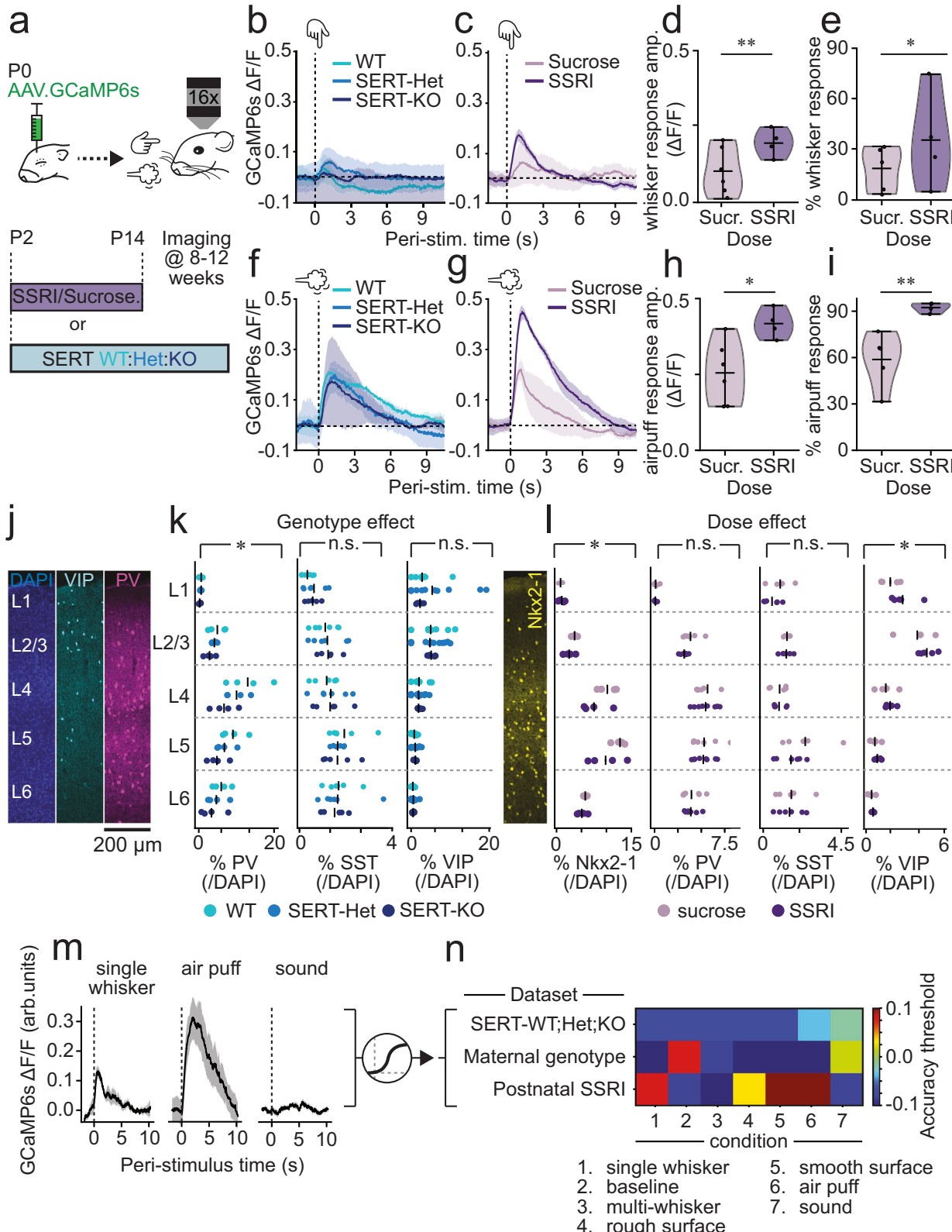

(S1BF) neocortex. Specifically, increased serotonin reduces the amplitude and duration of high synchrony H-events, while paradoxically increasing their frequency[6]. These observations likely reflect three non-exclusive mechanisms operating during early life. First, elevated serotonin in SERT-KO and SSRI-dosed pups likely attenuates sensory-evoked responses during the Th-SERT developmental window[26–29] via presynaptic 5-HT$_{1B}$ receptors expressed in

thalamocortical afferents. Second, elevated serotonin will also stimulate 5-HT$_{3A}$R-expressing interneurons, further restricting pyramidal cell activity[2,33]. Third, as we show here, altered serotonin impacts on the function of other GABAergic interneurons, including SST subtypes that are known to influence H-events in sensory cortex[5]. Direct investigation of PV interneuron is not possible given the delayed expression of this marker until the second postnatal week, following onset of the

**Fig. 7 | Postnatal SSRI exposure in wildtype pups and maternal SERT disruption lead to life-long changes in cortical encoding. a** Sensory stimulation in adult mice, either SERT-WT/Het/KO or wild-types postnatally treated with sucrose/SSRI. **b,c,** GCaMP6s response to single-whisker stimulus in (**b**) SERT-WT/Het/KO or (**c**) sucrose/SSRI-treated mice. **d,e,** Violin plots of single-whisker stimulus response amplitude (**d**, Mann–Whitney $U$-test $p = 0.004$) and cell recruitment (**e**, $t$-test $p = 0.032$) in sucrose/SSRI-treated mice. GCaMP6s response to airpuff in (**f**) SERT-WT/Het/KO or (**g**) sucrose/SSRI-postnatally treated mice. Violin plots of airpuff response amplitude (**h**, $t$-test $p = 0.046$) and cell recruitment (**i**, $t$-test $p = 0.007$) in sucrose/SSRI-treated mice. **j** Confocal image of a S1BF cortical column (200 μm wide) on an adult WT cortex, illustrating DAPI (blue, left), VIP (cyan, centre left) and PV (magenta, centre right) distribution. **k** Distribution of PV (Two-way ANOVA: genotype $p = 0.004$, layer $p < 0.001$, genotype-layer interaction = 0.625), SST (Two-way ANOVA: genotype $p = 0.934$, layer $p < 0.001$, genotype-layer interaction = 0.971) and VIP (Two-way ANOVA: genotype $p = 0.766$, layer $p < 0.001$, genotype-

layer interaction = 0.946) interneurons across layers for SERT-WT/Het/KO mice. **l** Distribution of Nkx2-1 (Two-way ANOVA: treatment $p = 0.032$, layer $p < 0.001$, treatment-layer interaction = 0.789), PV (Two-way ANOVA: treatment $p = 0.322$, layer $p < 0.001$, treatment-layer interaction = 0.903), SST (Two-way ANOVA: treatment $p = 0.138$, layer $p = 0.01$, treatment-layer interaction = 0.696) and VIP (Two-way ANOVA: treatment $p = 0.032$, layer $p < 0.001$, treatment-layer interaction = 0.691) interneurons across layers for sucrose/SSRI-treated mice. **m** Post-stimulus calcium responses to different stimuli were used for logistic regression analysis. **n** Heat map with the decoding accuracy of the different logistic regression classifiers; colours indicate accuracy minus threshold for significance of the classification as defined by bootstrapping. Positive values indicate significant classification accuracy. Sample sizes are 10 (WT), 8 (SERT-Het), 5 (SERT-KO), 6 (Sucrose) and 4 (SSRI). Traces are presented as mean ± SEM. Histology scatter plots are shown as individual animals (dots) plus the mean (line). Violin plots are presented as mean values ± max/min values.

Adult SERT expression pattern. We—similar to others[48]—set out to use a subtractive approach to resolve the impact of elevated serotonin on PV subtypes. However, detailed analysis of our experimental parameters suggests that our minimal stimulation optogenetic strategy can effectively isolate the contribution of PV cells ex vivo. This mirrors our recent findings, where we found that optotagging of Nkx2-1 interneurons identified a feed-forward inhibitory component in vivo[69] and other studies[70] where they report optical activation of ChR2-expressing PV interneurons at lower power intensities than that required for SST interneurons.

The most studied SERT polymorphism is a 44-base pair insertion/deletion in the gene promoter, leading to the short (S) and long (L) alleles[71,72]. The S allele reduces transcriptional efficiency, resulting in lower SERT expression and elevated extracellular serotonin levels[73,74]. Clinically, the S allele is associated with an increased risk for neuropsychiatric conditions such as depression[34,72,75,76], PTSD[36], and eating disorders[77], particularly when combined with environmental stressors[34,37,78–80]. However, some studies have failed to replicate some of these findings[81,82]. In our study, SERT-Het mice, which partially model the reduced SERT expression seen in S allele carriers, exhibit reduced H-event amplitude and increased frequency, with no significant changes in H-event duration. These features are distinct from the more pronounced changes observed in SERT-KO mice, including decreased H-event duration and increased VIP interneuron amplitude, which are absent in SERT-Het mice. Interestingly, SERT-Het mice more closely resemble SSRI-treated animals during the Th-SERT period, which shows hypoactivity and reduced H-event amplitude without duration changes or VIP amplitude increases, likely reflecting the partial blockade of SERT. However, some differences were observed between SSRI and SERT-Het/KO, such as increased Nkx2-1 interneuron responsivity during the Th-SERT time period in the former or increased VIP H-event amplitude in the later, which highlights the differential effects of life-long genetic versus postnatal pharmacological disruptions of SERT. Future studies should explore interactions between SERT-Het genotype and SSRI exposure as a model to investigate combined genetic and environmental contributions to cortical development. It is worth further noting that the same developmental effects on serotonin and calcium dynamics were observed in both male and female mice (Supplementary Fig. S11), suggesting that this phenotype is common to both, albeit direct comparison was precluded by sample sizes.

Thus, from a pre-clinical perspective, these results characterise the contribution that altered serotonin signalling plays as a risk factor for psychiatric neurodevelopmental disorders, be that as a result of *SERT* polymorphisms[35–38,83] or developmental exposure to SSRIs[41,84]. These results provide a unified conceptual framework in which to interpret recent studies relating disorders such as ASD with alterations in the integration of bottom-up and top-down pathways[85,86] as well as

excitation-inhibition balance[87–89]. Further, they are in accord with pathophysiology of sensory hypersensitivity, present in neurodevelopmental disorders such as ASD[90], again, associated with altered GABAergic signalling[91–93]. We speculate that postnatal serotonergic responses to aversive stimuli might provide the neurophysiological basis for the previously suggested interaction of SERT disruptions and early exposure to trauma[34,37,75]. In addition, it is worth noting that we report three distinct developmental trajectories to serotonin disruption: SERT-Het maternal genotype that alters connectivity in layers 2/3, SERT-KO, which results in hyperexcitability during early life but not adulthood, and postnatal SSRI dosing, which resulted in a prolonged effect into adulthood. The latter supports observations[2,31,60] that find the immediate postnatal period in mice to be critically important for the development of cortical microcircuits and specific to the role of serotonin signalling.

In summary, our results suggest that environmental, pharmacological, and genetic disruptions of perinatal serotonin dynamics can alter cortical patterns of activity, thereby perturbing interneuron integration and function with consequences for cortical information transfer and sensory encoding throughout life.

## Methods

### Animal husbandry and use

Experiments were approved by the local ethical review committee at the University of Oxford and performed in accordance with UK Home Office project licences P861F9BB7, PE5B24716, and PP8136190 under the UK Animals (Scientific Procedures) 1986 Act. The following mouse lines were used (all maintained on a C57BL/6J background): *SERT*-mutant mice (Slc6a4(tm1Kpl/J)), *Nkx2-1-Cre* (Tg(Nkx2-1-cre)2Sand/J), *SST-ires-Cre* (Ssttm2.1(cre)Zjh/J), *Ai32* (Gt(ROSA)26Sortm32 (CAG-COP4*H134R/EYFP)Hze/J), *Ai9* (Cg-Gt(ROSA)26Sortm9(CAG-tdTomato)Hze/J), *Olig3-Cre* (Olig3tm1(cre)Ynka/J), *Dlx5/6-Cre* (Tg(mI56i-cre, EGFP)1 Kc/J) and conditional *Celsr3* (Celsr3^{fl/fl}) mice. Mice of either sex were generated via targeted breeding and housed in a temperature (20–24° Celsius) and humidity (40%)-controlled room under a 12 h dark/light cycle with free access to food and water *ad libitum*.

### Surgery

Neonatal injections: intracranial injections were performed at postnatal day 0–1, as in previous work[94]. In brief, mice were anaesthetised with 3.5% isoflurane (1 L/min) in an induction chamber, and were then kept under anaesthesia using a custom-designed nose cone[95]. With a Nanoject III injector (Drummond) and a motorised stereotaxic arm (MCI), animals were injected in S1BF (AP 1.4 ML 1.5 from lambda), penetrating through the skull (right hemisphere). Pups were injected with 300 nL of either AAV9.hSyn-g5-HT3.0 (WZBiosciences, titre $\geq 1 \times 10^{13}$ v.g./mL), and AAV1.hSyn.GCaMP6s.WPRE.SV40 (Addgene, titre $> 1 \times 10^{13}$ v.g./mL). Injections of 100 nL were performed

at a rate of 23 nL/s at 3 different depths from skull surface: 600, 450 and 300 µm. After ~20 s the capillary needle was extracted and the pups were placed on a Thermacage (Datesand technologies) for recovery. In order to remove surgical odours, pups were rubbed in nesting material before being returned to their dam.

Cranial windows: P6-7 mice were kept on a heating pad to avoid hypothermia and anaesthetised with 4% Isoflurane (1 L/min) in an induction chamber. Then, mice were placed in a face mask delivering isoflurane, the concentration was constantly adjusted to keep the mice in a safe anaesthetic depth (1–4%), and fixed with earbars in a stereotaxic frame. Following incision and partial removal of the scalp, a blue lamp and a GFP filter mounted on a miner helmet (BLS) were used to identify the region of GCaMP expression. g5-HT3.0 was generally not detectable through the skull and coordinates AP = 2.4, ML = 2.8 were used. Then, a manually thinned head-fixing plate (<0.3 g) was cemented (Super-bond, Dental Prestige) over the region of interest and once dry, the animal was head-fixed and a 3 mm craniotomy was performed over the injected area (~S1BF) using an insulin syringe. Once haemostasis was achieved, two coverslips (3 mm and 4 mm) glued together with optical adhesive (Norland) were placed over the craniotomy. Then, the craniotomy was sealed with Vetbond (3 M) and fixed with dental cement (Super-bond, Dental Prestige). Animals were allowed to recover for at least 30 min. on a heat pad before being returned to their home cage. No signs of abnormal skull or brain development were observed in these animals (Supplementary Fig. S9). Moreover, their developmental timeline (e.g., eye opening) was similar to animals that did not undergo the procedure (Supplementary Fig. S9b). Good cranial window visibility was achieved throughout development (Supplementary Fig. S9c).

### In vivo experiments

Postnatal drug delivery: pups were gently scruffed and sucrose (5 mg/kg of body weight) or fluoxetine (10 mg/kg of body weight) diluted in sucrose was delivered with a micropipette delicately introduced on the tip of the mouth. The dam was kept in a different cage during pup-treatment and the procedure was kept under 15 min, to avoid excessive maternal separations. Animals were dosed using a fresh drug dilution of 0.1 mg/ml, daily from P2 to P14, both included, maintaining a regular time of the day for dosage within the same litter. On surgery days, pups were dosed prior to surgery. During imaging days, each animal was dosed 5 min before the start of its first recording to keep time from dosing homogeneous across animals.

Two-photon imaging: recordings were performed on a resonant galvo scanning 2-photon microscope (Bruker) with a Chameleon Ultra II laser (Coherent) and 50 mW of power on sample. A 16x/0.8-NA water immersion objective lens (Nikon) was used. g5-HT3.0, and GCaMP6s were imaged using a 920 nm beam while tdTomato was imaged at 765 nm. Imaging was performed at a frame rate of 30 Hz in a square field of view (643 × 643 µm for GCaMP6s and 287 × 287 µm for g5-HT3.0). All recordings were obtained at ~150 µm from the brain surface (cortical layer 2/3). Upon visual inspection, animals with observable movement artefacts were not recorded. We observed high stability in these animals across development, likely driven by the brain expansion at this period, providing pressure against the window and subsequently increasing recording stability. Animals were imaged daily during a 20 min baseline session and a 12 min whisker stimuli delivery session from P7 to P16 for developmental imaging. On the 1st day of recordings, the field of view was chosen based on whisker-stimulation responsiveness. Then, the same field of view was found by matching the sparse patterns of interneurons at the start of the first recording every day (Supplementary Fig. S10d), ensuring the same region was recorded. Pups were surrounded by padding material to minimise heat loss and homecage nesting material was added to this padding to reduce stress by adding homecage odours. Adult imaging was performed in mice 8 to 12 weeks old, only when the window maintained

good visibility and excessively bright cells (potentially calcium-filled) were not present, or very sparsely so (<1% of total cells), in the field of view.

Sensory stimulation: sensory stimuli were generated pseudo-randomly using custom MATLAB code. Whisker stimulation was delivered with a piezoelectric actuator (Physik Instrumente) connected to a custom-designed (3D printed) whisker stimulator with modules for single-whisker (glass capillary), multi whisker (brush), airpuff (picospritzer III at 80 psi), smooth surface (velcro), or rough surface (sand paper). All whisker stimuli were delivered oscillating at 20 Hz within the mouse's left whiskers, with a duration of 40 ms. Stimuli were presented 10 times with inter-stimulus intervals between 20 and 30 s. The auditory stimulus was an amplitude and frequency modulated complex tone with a 5 kHz carrier frequency delivered with a speaker (Dell). All traces were time-locked using PackIO[96].

### g5-HT3.0 sensor data analysis

All images were pre-processed by registration with Turboreg[97] against a mean-intensity average of 200 frames in Fiji[98]. All fluorescent (F) traces are presented as ΔF/F:

$$\Delta F/F = \frac{(F - Fmean)}{Fmean} \tag{1}$$

where Fmean is the mean fluorescence of the 20 min baseline recording session. For stimulus responses in animals P14 or older, when a hemodynamic response was observable, tdTomato fluorescence was used as a ratiometric control, as in our previous work[15]. Signal filtering was performed using the Savitzky–Golay filter with a third order polynomial on windows of 31 frames (~1 s). For subregion analysis, custom-designed code in Fiji was used to quantify mean fluorescence in a 256 subregion grid.

### Calcium data analysis

We used Suite2p[99] to perform registration, cell detection and fluorescent trace extraction of all calcium recordings. Given the difficulty of finding the right correction factor under conditions of high cell density (Supplementary Fig. S10b) and very high cell-neuropil correlation (Supplementary Fig. S10a)[100], we did not perform neuropil subtraction. We confirmed that the raw GCaMP6s intensity values were not significantly altered by genotype or treatment condition across the different developmental recordings, which suggests that sensor expression levels were not biased towards any experimental group (Supplementary Fig. S10c). Regarding signal normalisation, due to the change in calcium neurophysiology across development, and the importance of the baselining strategy to compare measurements from different animals, we chose to normalise against the first percentile, as this showed the highest stability across development (Supplementary Fig. S10f). Thus, calcium developmental recordings are presented as ΔF/F:

$$\Delta F/F = \frac{F - F(1st\ percentile)}{F(1st\ percentile)} \tag{2}$$

where F(1st percentile) is the first percentile of the raw fluorescence values of each cell across the whole recording. Adult recordings were normalised against the mean as in our previous work[101].

High (H-) and low (L-) synchronicity developmental calcium events were detected with a custom-made algorithm optimised in control animals until good detection was achieved (Supplementary Fig. S10e), yielding similar event frequency to previous studies[6]. Given the duration of events is ~2–20 s (Fig. 2 and Supplementary Fig. S2), detection was performed in 6 s time bins from the 20 min baseline recording session. First, if the mean fluorescence across all cells was ever greater than the mean ΔF/F, the maximum was selected as the

event peak. Next, adjacent bins were searched to determine if any maximum reached in those bins exceeded the selected event peak in the current bin, implying that the event stretched across more than one time bin. The event peak was therefore defined as the maximal peak across the bin and adjacent bins. Event duration was defined as the period around the event peak where the mean $\Delta F/F$ of all cells was greater than the mean. Single cells were considered to participate in an event when their mean fluorescence within the event period was greater than the mean fluorescence of the individual cell. H-events were defined as events with more than 80% of cells participating, and L-events as 20–80% of cells being recruited, based on previous studies[5,6].

Automatic interneuron detection based on tdTomato expression was optimised to minimise false positives, allowing identification of a subset of interneurons within the field of view. Cells were identified as interneurons when their mean fluorescence during a 20-s 765 nm-laser recording was equal or higher than the 97th percentile.

Correlations were calculated between all cell pairs using Pearson's correlation coefficient. For correlations across distance, suite2p ROIs coordinates were used to calculate the Euclidean distance between cells and the average cell-to-cell correlation was obtained on bins of ~40 μm distance. Whisker response amplitude was defined as the maximum $\Delta F/F$ value on a 20 s post-stimulus window of time (i.e., within the interstimulus interval). Cell-neuropil correlations were calculated with Pearson's correlation coefficient between the entire trace from the 20 min baseline session of the suite2p-extracted ROI and its surrounding neuropil region.

Neuronal recruitment upon sensory stimuli was determined by calculating the proportion of responsive neurons. Responsiveness was defined as statistically significant differences between the 1 s mean pre-versus post-stimulus onset with paired $t$-test or Wilcoxon signed-rank test after addressing normality with Shapiro–Wilk test and correcting for multiple comparisons with Benjamini/Hochberg method.

Genotype logistic regression classifiers of Fig. 5 used a logistic regression model with an L2 penalty term to the weights with a regularisation strength of 0.001 (after parameter optimisation). The Scikit-learn implementation of logistic regression was used[102] on average fluorescence of 500 ms time bins for 500 randomly selected cells, on a 20 s post-stimulus window of time, creating 500-element feature vectors. A classifier was trained for each stimulus type and time bin. A 60/40 random train/test split was performed across individual animals to ensure subject-wise cross-validation[103]. Test decoding accuracy was calculated for each time bin and the maximum decoding accuracy was selected (Fig. 7o). The accuracy of 7000 classifiers trained in permuted data was used to define significance thresholds (Supplementary Fig. S6) for each classifier. Logistic regression classifiers for stimulus type (Supplementary Fig. S6d) were implemented without time bins or averaging, but by using a 20 s post-stimulus window (600 frames) of 500 randomly selected cells to create a 300,000-feature vector. A single classifier was generated for each animal, and cross-validation was performed recording-wise.

## Behavioural tracking analysis

Time-locked recordings at ~30 frames/s of the mice while performing two-photon imaging were obtained with a camera (point grey CM3-U3-13Y3M-CS) and an infra-red light (BW 48 LED). Movies were then used to train a ResNet-50 artificial neural network with 200 manually-labelled frames from 20 recordings, extracted with K-means clustering, to track 3 markers (nose, left and right forelimbs) using DeepLabCut[104]. Successful tracking was achieved after 500,000 epochs and pose estimation was performed for all recordings. Movement was calculated as the Euclidean distance between the two-dimensional coordinates of consecutive frames for each marker. Sleep state was estimated from the left forelimb (i.e., the marker with best visibility) movement with the following criteria: continuous movement

was considered wakefulness, lack of movement quiet sleep and sharp twitches active sleep.

## Ex vivo electrophysiology

Preparation of brain slices and laser-scanning photostimulation (LSPS) was performed as previously described[51]. Following anaesthesia with 4% isoflurane in 100% $O_2$, mice were decapitated and brains removed in ice-cold artificial cerebrospinal fluid (ACSF containing, in mM, 125 NaCl, 2.5 KCl, 25 NaHCO$_3$, 1.25 NaH$_2$PO$_4$, 1 MgCl$_2$, 2 CaCl$_2$, 20 glucose; pH equilibrated with 95%$O_2$/5% $CO_2$). In the same ACSF solution, 350 μm coronal slices containing S1BF were taken using a vibratome and maintained in room-temperature ACSF for 1 h before transferring to the recording rig. Individual cells were visually selected on-screen for patch clamp using a 40x water immersion objective with an Axio-Cam MRm camera attached to Zeiss Axioskop 2 FS plus microscope (both Zeiss Ltd, UK), and whole-cell recordings made at room temperature using a Multiclamp 700B amplifier and Digidata 1440 A digitiser (Molecular Devices, USA) with data sampling at 20 kHz. Cells were patched with borosilicate glass pipettes pulled using a Narishige PC-10 microelectrode puller to a resistance of 6–8 MΩ and filled with caesium-based solution containing, in mM, 110 gluconic acid, 40 HEPES, 5 MgCl$_2$, 0.2 EGTA, 5 QX-314, 0.3 GTP and 2 ATP.

## Laser-scanning photostimulation (LSPS)

Prior to photostimulation, slices were maintained for a minimum of 6 min in high-divalent cation (HDC) ACSF of the same composition as the recording ACSF but with higher concentration (4 mM) of MgCl$_2$ and CaCl$_2$ and MNI-caged glutamate (200 μM; Tocris Bioscience, UK). UV laser pulses (355 nm; DPSL-355/30, Rapp OptoElectronic, Germany) were directed to the slices through a galvanometer scanner system (UGA-42; Rapp OptoElectronic) and focused through a 10X objective. To measure GABAergic inputs, laser-evoked inhibitory postsynaptic currents were recorded by clamping the cell at the reversal potential for glutamate ($E_{Glut}$) determined empirically. The LSPS grid was characterised by 11 × 17 spots stimulated in a pseudorandom sequence and spaced ~50 μm; long-duration (100 ms) laser pulses were administered at 1 Hz with the laser intensity calibrated to elicit action potentials only when targeting the soma of cells[105]. A minimum of three sequences were recorded in voltage clamp configuration and average charge of synaptic inputs from each 50 μm grid point was analysed. Analysis of LSPS data was performed with a customised MATLAB (R2020a, MathWorks) toolbox. Maps were normalised by dividing the input from each laser position by the sum of all pixels. Columnar profiles were obtained by summing all values for each line in individual heat-maps; layer distribution of inputs was calculated by summing all inputs occurring in all pixels belonging to a given cortical layer, with the layer boundaries manually determined using a DIC image acquired under 10X magnification. To generate average maps, individual maps were aligned at the base of the L4–L5 boundary.

## Optogenetics

Ex vivo optogenetics experiments were performed using a 470 nm light-emitting diode (LED; Thorlabs Inc., US or CoolLed, UK), focused onto the recorded neuron using a 40X objective. Light-evoked IPSCs were recorded voltage-clamping the target cell at $E_{Glut}$, whereas EPSCs were recorded near RMP (Vh = −60 mV). Brief LED pulses (0.2–30 ms) were used to activate cells expressing channelrhodopsin-2 (ChR2) and initiate post-synaptic events in the recorded cell; an inter-stimulus interval of 10–20 s was used to allow for ChR2 recovery. LED power intensity and stimulus duration were modified to determine the minimum postsynaptic current that could be elicited in recorded neurons. A minimum of 6 trials were recorded for the minimum and maximum postsynaptic currents that could be measured. Analysis of in vitro optogenetics data was carried out with customised scripts in MATLAB (Mathworks, USA). A large detection window for EPSC/IPSC

onset (0–50 ms from LED stimulus onset) was used to account for possible developmental effects on ChR2. For events within the monosynaptic detection window, synaptic event features were measured, such as peak amplitude, area under the curve (charge) and onset latency.

## Immunohistochemistry

Mice were anaesthetised in an induction chamber with isoflurane (5%, 1 L/min) and then injected intraperitoneally with a non-recovery dose of pentobarbital (Pentoject, Animalcare Ltd) (adjusted by mouse weight). The mice received intracardiac perfusion with 0.01 M PBS followed by 4% PFA (Sigma-Aldrich; 16% stock) for tissue fixation. The brain was dissected and left in 4% PFA for 24 h at 4 °C. Brains were placed in 0.01 M PBS, embedded in agarose 5%, and sliced using a vibrating microtome (Leica VT1000S). Coronal sections of 50 μm were made. Brain slices were conserved in 0.01 M PBS with 0.05% sodium azide, at 4 °C. Three brain sections from each animal were placed for 2 h in 2% goat serum in PBS-T (0.2% Triton in PBS) at room temperature. The tissue was incubated at 4 °C overnight with a primary antibody in blocking solution: rabbit anti-PV (1:1000) (PV27a, Swant, Switzerland; lot# AB2631173), rabbit anti-SERT (1:500) (Millipore Corp., USA; PC177L, lot# 3996287). Sections were washed with 0.01 M PBS 3 times and incubated for 2 h with the following secondary antibodies (1:400): goat anti-Rabbit-Alexa647 (AB_2535813) or goat anti-Rabbit-Alexa568 (Invitrogen A11011, lots# 2273773; 2782620) at room temperature. Tissue was washed with PBS and incubated with 1 ml of DAPI (1:1000) for 30 min. Finally, samples were washed with PBS again and mounted onto superfrost slides (ThermoScientific) with 0.01 M PBS. A cover slide was placed on top of the slide and sealed with nail polish.

All imaging and analysis were performed blind to the sample genotype/treatment. Histological imaging was conducted with an epifluorescence microscope (Zeiss) for injection site confirmation and imaging of SERT and a confocal microscope (Olympus Fluoview FV1000) for interneuron imaging. Imaging settings were kept constant within the experimental batch (SERT-KO: WT, HET, KO and SSRI: Fluoxetine and sucrose). In both cases, z-stacks of the whole slice depth were obtained (-11 images per slice). Three brain slices from each animal across both hemispheres were imaged.

For injection site confirmation, GCaMP6s fluorescence was visually confirmed in S1BF. For interneuron quantification, z-stacks were collapsed into z-projections using the max (SERT-KOs) or the mean (Fluoxetine/sucrose) with Fiji (ImageJ). Using QuPath[106], annotations were drawn over several 200 μm wide cortical columns (-5 per image). Sub-annotations were manually drawn in the columns for each layer, based on DAPI cytoarchitecture (Supplementary Fig. S8). Automatic cell detection was done with the DAPI channel in Qupath by optimising signal intensity, nuclei expected size, and cytoplasmic expansion thresholds. Interneuron classification was performed in QuPath[106] by training random tree classifiers in a subset of images until human-like classification performance was achieved. A classifier was trained on each marker (VIP, Nkx2-1 and PV). Automatic classification was run throughout annotations and images with custom-made workflow scripts. Counts for each layer were averaged across columns, hemispheres, and slices for each animal, such that all plotting and statistics were done using animals as independent data points.

## Statistics

Statistical analyses were conducted using Python 3.9 and all were two-sided tests unless otherwise indicated. Normality was assessed by the Shapiro–Wilk test. Two-group comparisons were performed with *t*-test/paired *t*-test and Mann–Whitney *U*-test/Wilcoxon signed-rank test for parametric and non-parametric tests of measurements in the same/different mice, respectively. Multigroup testing across a single variable was performed with either one-way ANOVA or Kruskal–Wallis test for parametric and non-parametric testing, respectively. Multigroup testing across two variables (e.g., across genotype and age) was performed by two-way ANOVA for normally distributed data, and two-way permANOVA, i.e., with permutation testing, for non-normally distributed data. Pairwise comparisons were performed with Fisher's Least Significant Difference (LSD) for three groups, Tukey's HSD test for more than three groups when the data was parametric and Dunn's test for non-parametric data. Multiple comparison correction was performed with Bonferroni-Holm correction unless otherwise indicated. Kolmogorov–Smirnov test was used to compare distributions. Bootstrapping was used to test the significance of logistic regression classifiers. All statistical analysis was performed using single animals as independent points, except when population analysis was performed (interneuron analysis), in which case, cells were used as independent points, but similar trends were confirmed when averaging by animals. Statistical significance was considered for $*p < 0.05$ and $**p < 0.01$.

## Software

Microscopy images were processed with Fiji[98] (processing package based on ImageJ version 1.54). Automatic cell detection and classification were performed with Qupath (v0.5.0)[106]. Turboreg plugin[97] was used for g5-HT3.0 image registration and Suite2p (v.0.14.0) for GCaMP6s recording registration, cell detection and signal extraction[99]. DeepLabCut (v2.3.5 and later) was used for behavioural tracking[104]. Ex vivo electrophysiology used pClamp 10 (Molecular Devices, USA). Figures were created from panels using Adobe Illustrator (v 29.3.1). All in vivo *and* ex vivo/in vitro analysis, statistics and plotting were performed in Python v3.9, while ex vivo electrophysiology was analysed in Matlab (R2020a, version 9.8 and later), with plotting and statistics performed in Prism v9 and v10 (GraphPad).

## Reporting summary

Further information on research design is available in the Nature Portfolio Reporting Summary linked to this article.

# Data availability

Source data are provided with this paper. Processed data is available via the University of Oxford research archive (ora.ox.ac.uk) with specific datasets available by contacting the corresponding author. Source data are provided with this paper.

# Code availability

All code used for data analysis is available either by contacting the corresponding author or accessing the Butt lab Github (https://github.com/Butt-lab).

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

## Acknowledgements

This project was funded by the Medical Research Council (Grant MR/T033320/1 to S.J.B.B. and T.S.), the Marie Skłodowska-Curie Action Serotonin and Beyond under the European Union's Horizon 2020 research and innovation programme (Grant Agreement 953327 to T.S.), and the Wellcome Trust (204651/Z/16/Z) and the European Research Council (ERC) under the European Union's Horizon 2020 research and innovation programme (grant agreement No. 852765) to A.M.P. We would like to thank Nina Soto and the Bacci lab for providing expertise on the SSRI oral dosing paradigm. We would also like to thank the following for sharing the transgenic mouse lines used in this study: Prof. Klaus-Peter Lesch (conditional SERT line), Yasushi Nakagawa (Olig3-Cre), Andre Goffinet and Fadel Tissir (conditional Celsr3).

## Author contributions

G.O.S., H.W., A.M.P., T.S., and S.J.B.B. conceived and designed the experiment. G.O.S., H.W., V.M., H.M., C.G., C.H., J.A.S., A.S., I.P.L., A.B., S.J.B.B. performed the experiments. G.O.S., H.W., V.M., H.M., C.G., C.H., J.A.S., and A.S. analysed the data. F.D. and Y.L. shared viral constructs. A.M.P. provided technical expertise. G.O.S., H.W., and S.J.B.B. wrote the manuscript with input from A.M.P. and T.S.

## Competing interests

The authors declare no competing interests or other interests that might be perceived to influence the interpretation of the article.

## Additional information

**Gabriel Ocana-Santero**[1,2], **Hannah Warming**[1], **Veronica Munday**[1], **Heather A. MacKay**[1], **Caius Gibeily** ⓘ [1], **Christopher Hemingway**[1], **Jacqueline A. Stacey**[1], **Abhishek Saha** ⓘ [1], **Ivan P. Lazarte**[1], **Anjali Bachetta**[1], **Fei Deng** ⓘ [3,4], **Yulong Li** ⓘ [3,4], **Adam M. Packer** ⓘ [1], **Trevor Sharp** ⓘ [2] & **Simon. J. B. Butt** ⓘ [1] ✉

[1]Dept. of Physiology, Anatomy and Genetics, Oxford University, Oxford, UK. [2]Dept. of Pharmacology, Oxford University, Oxford, UK. [3]State Key Laboratory of Membrane Biology, Peking University School of Life Sciences, Beijing, China. [4]PKU-IDG/McGovern Institute for Brain Research, Beijing, China. ✉e-mail: simon.butt@dpag.ox.ac.uk

