## [Transparent Peer Review file · Nature Communications]

Perinatal serotonin signalling dynamically influences the development of cortical GABAergic circuits with consequences for lifelong sensory encoding

Corresponding Author: Professor Simon Butt

Version 0:

Reviewer comments:

Reviewer #2

(Remarks to the Author)

By using *in vivo* longitudinal imaging of both serotonin signaling (g5-HT3.0) and neuronal activity (GCaMP6s), Ocana-Santero and colleagues examined serotonergic activity, cortical activity and local interneuron activity in somatosensory cortex (S1BF) of both genetic and pharmacological mouse models that have altered serotonin levels across brain development. This manuscript reports that cortical serotonin levels are dynamic with behavioral state early in development and that serotonin represents an important signal in response to aversive stimuli. The authors show that transient expression of the serotonin reuptake transporter, SERT, clamps serotonin responses to sensory-evoked activity during early development. The prevention of this serotonin “buffering” by either genetic or pharmacological manipulation results in the developmental dysregulation of inhibitory circuitry resulting in cortical hyperexcitability; which in turn has long-lasting effects in cortical function *in vivo*.

This research is one of the few studies in the field that shows how serotonin regulates cellular and circuit plasticity in cortex across development using elegant *in vivo* approaches. This study will also generate interest to the general audience in the neuroscience community. However, there are several important loose ends the authors should address and additional control experiments are needed. To strengthen their findings, the authors should address the concerns listed below:

Major comments:

1. How were the Th-SERT and Adult SERT periods validated for this study? Figure S1 does not show developmental changes in SERT expression across their experimental time window. The authors should also verify the functional consequences of Th-SERT (< P10) on 5HT levels in S1BF by using their sensors combined with optogenetics or chemogenetics. Increased signals of g5-HT3.0 in later development during Adult-SERT (or lower signals in Th-SERT stage, in figure 1b and c) might simply reflect the different activity of 5HTergic neurons across development, different amount of 5HT release between Th-SERT and Adult-SERT stages, and/or different expression levels of AAV-g5-HT3.0 in S1BF. The buffering effect by Th-SERT should be experimentally confirmed.
2. In figure 2, the authors make a case for cortical hypoactivity based on the relative change in fluorescence. However, the authors also report an increase in H-even frequency that was accompanied with the decrease in amplitude and duration. This “increase” in frequency suggests an enhanced neuronal activity rather than an overall decrease in activity. Where and how does the increase in frequency fit within the hypoactivity model?
3. In figure 3, it is unclear why the authors did not probe potential changes in PV's inputs to PYR. This experiment will be essential for and strengthen the authors' argument since the conclusion is that SST's input to PYR is increasing to compensate for PV's decreased input, yet PV interneurons were never examined. How does elevated serotonin in SERT-KO pups impact the early development of PV?
4. In figure 3, the authors should measure mIPSCs on L2/3 pyramidal neurons from each genotype (WT, Het, KO) to confirm that the changes in ChR2-evoked IPSCs are not due to alterations of inhibitory post-synapses on pyramidal neurons in Het

or KO condition (2c, d and e). Inhibitory synapse development (post-synapse number and/or strength) could have been affected in SERT-KO mice.

5. In figure 4, does postnatal fluoxetine treatment increase serotonin levels in the cortex? If it does, is this increase diminished during the Th-SERT period, but not Adult-SERT stage? This will be very important evidence supporting the buffering role of Th-SERT in cortex.

6. In figure 4i, sucrose treated mice showed substantial reduction in GCaMP6s signals, which could have led to a significant difference in GCaMP6s comparison between SSRI and sucrose treated groups. This reduction was not observed in figure 2j (I assume the same experimental conditions with the same ages). Control (sucrose) data must be well validated first to determine hypo vs hyper-excitability.

7. In figure 5l and m, the authors should show % SST, % PV, and % Nkx2-1 together.

8. The figures are very difficult to follow without simultaneously reading the text together with the figure legends. The authors sometimes add the postnatal age of the mice above the figure and sometimes it is just referenced in the text, or sometimes in the figure legends. For example, figure 2 e-h are from P7-10. Figure 2 j is from P11-13. Figure 2 m and o are from P7-10. Also, sometime Th-SERT is indicated on a figure panel and sometimes not. Please show Th-SERT vs Adult-SERT for each panel, when applicable. Young vs old and Th-SERT vs Adult-SERT are the key features of this study.

9. The entire discussion needs to be expanded significantly. The authors should elaborate on what they think about how serotonin signaling is involved in pyramidal neuron activity and interneuron activity in the somatosensory cortex, and why their activity is so special during development. In addition, more about SERT polymorphisms should be added since Th-SERT vs Adult-SERT represents a large component of the rationale for the research work. Additionally, the authors should clarify the relevance of SERT-Het and SERT-KO relative to the polymorphisms seen in clinically relevant populations, especially since different responses are observed in Het vs KO in some datasets (e.g., figure 2n)

Minor comments:

1. In figure 1 d and e, serotonin levels were measured near VIP (subregions surrounding VIP). Please accurately define the "proximal" in the text. Did the authors observe similar results if they examine the signals only from VIP-INs? Also, in figure 1 d and e, please clearly indicate ROIs of their measurement. The images are very vague.

2. In figure 1 g, h, and i, are g5-HT3.0 data from WT during Th-SERT period? Please label the genotype and experimental stage.

3. In figure 2 n and p, please show time-locked changes in GCaMP6s for whisker stim experimental data across development. At least, time-locked changes of VIP from Th-SERT vs Adult-SERT similar to data in figure j. Statistical comparisons are not clear on figure 2 n and p. Please also show H-event duration from VIP (Figure 2 m) and Nkx2-1 (Figure 2 o) of each genotype.

4. In figure 3 k, n, and o, please clearly label if WT and HET indicate the maternal genotype or pups' genotype.

5. In figure 5 a-e, SSRI treatment period should be clearly shown in an experimental schema and/or timeline.

6. On line 206, please define SERTKO-HOM.

7. On line 220, it should be l, not i.

Reviewer #3

(Remarks to the Author)

This is an interesting and generally well executed study which examines the impact of genetics, pharmacology and aversive experiences on serotonin signaling and sensory processing. The authors used in vivo imaging of both serotonin signaling (using the fluorescent serotonin sensor g5-HT3.0) and neuronal activity (using GCaMP6) in genetic (SERT Het and SERT KO) and pharmacological (SSRI) mouse models. Combined with slice electrophysiology, they show that genetic deletion of the serotonin transporter or the administration of antidepressants lead to a shift from reduced cortical activity to hyperexcitable sensory cortex which results from changes in the cortical GABAergic microcircuitry. Specifically, they found that disruptions in early buffering led to altered development of interneuron subtypes of the cortical microcircuitry. Finally, they report that only postnatal exposure to antidepressants affects sensory encoding in adult mice since the same effect is not observed in SERT KO mice.

Although the story is compelling, a more cautious presentation would improve the paper. There are a few points where more careful consideration is needed:

Introduction: It would be useful to provide a couple of sentences and references explaining the developmental changes that the serotonergic system is undergoing according to age. For example, the authors report that P11 to P13 reflect adult-like SERT expression. Isn't this expression brain area dependent? Please provide the necessary explanations and literature.

Results:

1. Line 80: The serotonin sensor g5-HT3.0 was injected into the somatosensory cortex of which mice? Do the authors mean the VIP-Cre;Ai9? This information should be clarified from the beginning of the paragraph and the authors should explain already from the beginning why they chose this mouse line to test serotonin signaling to help with the flow of the story.

Line 93: How do the authors define localized signaling and what the authors consider as proximal to VIP interneurons? A few details should be provided here concerning the responses proximal to VIP.

2. It would be nice to see a Time-series video showing serotonin fluctuations. For example, the authors could provide along with the Extended Data Fig. 1a the corresponding video.

How many mice were tested? Please provide the number of animals for each graph.

3. The authors suggest that SERTKO mice transition from cortical hypoactivity to hyperexcitability. They show that during the early postnatal time period KO mice show a decrease in the amplitude and duration of H-events (Fig. 2e and f). However, the mean H-event frequency in KO mice is higher than WT (Extended Data Fig. 2b) at P7-10. This is a rather important information that should be moved to the main figure. How do the authors explain this increase in frequency and how this is aligned with hypoactivity? What is the mean frequency of H-event in VIP mice for WT, Het and KO? The mean frequencies should be plotted for all conditions for VIP and Nkx2-1 mice.

Images in b and l should become bigger. The cells are not visible, especially for the Nlkk2.

4. It would be useful to see representative rasterplots for a given population of simultaneously recorded neurons to appreciate the presence of L- and H-events. These graphs should be added along with the heatmaps in Fig. 2c and d.

5. The recordings of SST interneuron inputs in Fig. 3 show an increase in the conductance of IPSCs in SERT-KO L2/3 pyramidal neurons. The authors suggest that elevated serotonin levels in SERT-KO pups impair the early development of PV interneurons and this impairment is so severe that the observed increase in SST interneuron input cannot compensate, leading to hyperexcitability in L2/3 of the S1BF around the onset of active sensation. Recordings of PV interneuron inputs are essential to corroborate this hypothesis in WT, Het and KO mice.

6. The VIP amplitude during H-event after SSRI dosing is not altered (Fig. 4 and Extended Fig. 4). How do the authors explain this difference in comparison to the strong effect they found for the KO mice?

The number of recorded neurons should be indicated along with the number of mice.

7. The authors show that postnatal SSRI administration results in long-lasting effects on sensory encoding. They show that single-whisker stimulation or airpuff presentation increased cortical response amplitude and neuronal recruitment in SSRI treated animals but not in the KO and they hypothesize that this is due to a prolonged impairment of the inhibitory control of information transfer. They present stainings for VIP and PV interneurons. SST quantifications should be added in a similar manner for WT, Het and KO mice since these neurons have a critical role in cortical circuitry as it was also suggested in Fig. 3 to support their results.

8. It will be nice to see analysis of synchronous events in the adult mice of 8-12 weeks that are presented in Fig. 5 to evaluate possible alterations that may occur postnatally in later ages as a result of SSRI exposure and maternal SERT disruption. This figure could be added to the supplementary material of the paper for the different conditions of single whisker deflection and airpuff stimulation.

9. Did the authors see any differences on serotonin signaling among male and female mice? It would be very interesting to see if the dynamics change in relation to the sex of the mouse since they authors used both sexes for their study. If this is not possible, at least it should be discussed in the discussion paragraph.

Other comments:

10. Why did the authors image tdTom at 765nm (line 578)? With an absorption peak around 554nm and emission peak around 581, one would expect that for 2P imaging the authors would image tdTom above 1000 nm.

Line 603: I would suggest to replace Fmean by F0 to avoid confusions. Otherwise, the authors should clarify at line 604 that Fmean is the mean baseline fluorescence of the recording session.

11. How did the authors account for movement in the z-plane for the analysis of the two-photon calcium imaging data? A description should be added to the Methods part.

12. For all figures, please provide the number of cells and number of mice recorded and analyzed.

Version 1:

Reviewer comments:

Reviewer #2

(Remarks to the Author)

The authors have properly addressed most of my concerns by adding new experiments and editing the text. It is a beautiful work.

However, I still have doubts if the authors can claim on their ms that the SERT “transiently” expresses during early development “without” showing the “transient” expression of SERT in their system. The references they cited are good and sufficient for making a rationale for their work, but not for demonstrating that the SERT is indeed expressed transiently in S1BF.

Fig S5 A might indirectly support it though. What is the age of slice on the panel B of Fig S1?

I would show the similar images of SERT expression from Th-SERT (I assume the current Fig S1 B) and Adult-SERT, side by side, in order to claim that SERT expresses “transiently” in S1BF.

Minor points

1. * on VIP of Fig1 D might be an error. Its legend indicates $p = 0.007$ (line 155).

Line 173: * $P < 0.05$, ** $P < 0.05$ seems to be a typo.

2. Figure 1 B and C, airpuff (black) and whisker (dark gray) are clearly indicated, but the data point/line for sound (light gray) is not clearly shown.

3. Figure 2, ANOVA (lines 216, 223, 226) needs to be clarified. One way or Two way.

4. Figure 3 A, labeling (or color-coded) bottom-up, feed-forward, and top-down inhibition will be helpful for readers.

5. Line 342: not h but j.

6. Figure 4 M, showing HET data will be useful for comparison.

7. The authors could add “in WT” on Figure 6 title. For example, Postnatal fluoxetine treatment in WT recreates the SERT-KO transition from cortical hypoactivity to hyperexcitability.

Reviewer #3

(Remarks to the Author)

The authors have significantly revised the manuscript by performing additional analyses and incorporating more data into the main figures, including additional Extended Data Figures. Furthermore, they have substantially enhanced the discussion section. I have no additional comments.

Version 2:

Reviewer comments:

Reviewer #2

(Remarks to the Author)

It is an outstanding work.

Response to Reviewers - Nature Communications

Manuscript ID: NCOMMS-24-37009-T

We would like to thank the reviewers for taking the time to read the manuscript and for providing positive feedback that has greatly improved our manuscript.

Reviewer 1

Major comments:

1. How were the Th-SERT and Adult SERT periods validated for this study? Figure S1 does not show developmental changes in SERT expression across their experimental time window. The authors should also verify the functional consequences of Th-SERT (< P10) on 5HT levels in S1BF by using their sensors combined with optogenetics or chemogenetics. Increased signals of g5-HT3.0 in later development during Adult-SERT (or lower signals in Th-SERT stage, in figure 1b and c) might simply reflect the different activity of 5HTergic neurons across development, different amount of 5HT release between Th-SERT and Adult-SERT stages, and/or different expression levels of AAV-g5-HT3.0 in S1BF. The buffering effect by Th-SERT should be experimentally confirmed.

- We define the Th-SERT and Adult SERT time periods based on current literature and apologize for omitting the relevant citations in our original submission. A number of studies report that transient SERT expression in the ventral posteromedial nucleus of thalamus is present between embryonic day (E)15 and postnatal day (P)10 (see references 18, 29-32). We have updated the introduction (page 1, lines 59-63) to highlight these previous findings and provide additional context for our study.
- The reviewer suggests that the absence of serotonin signalling to whisker and auditory stimuli during the Th-SERT period (**Fig. 1b,c**) could be, in part, explained by low levels of g5-HT3.0 expression at early ages or differences in serotonergic activity. We suggest that GRAB expression levels are unlikely to be a contributing factor as this would be at odds with our observations from both our genetic model - the SERT knockout pups, and those dosed acutely with SSRIs. In both cases, we observe 5-HT signalling in response to whisker stimulation during this early time period using the GRAB sensor (**Fig. 1f-h**). Moreover, the presence of a robust 5-HT signal to the aversive airpuff stimulus (**Fig.1b**) - on a par with that seen in adult SERT (**Fig. 1c**), suggests that it is possible to overcome buffering of the 5-HT signal under certain, more extreme circumstances. We have reworded the text on **page 3** (lines 107-112) in light of the reviewer's comment.

2. In figure 2, the authors make a case for cortical hypoactivity based on the relative change in fluorescence. However, the authors also report an increase in H-event frequency that was accompanied with the decrease in amplitude and duration. This "increase" in frequency suggests an enhanced neuronal activity rather than an overall

decrease in activity. Where and how does the increase in frequency fit within the hypoactivity model?

- We thank the reviewer for raising this point. Our original presentation of the data was perhaps confusing and as a consequence we have remodelled **Fig. 2** to better capture our observations, for example, H-event frequency has now been included in the main figure (**Fig. 2f**). We agree with their assessment that at first glance these findings would appear to be contradictory but our observations are perhaps best visualised in the new representative data - raster plots in **Fig. 2b** (as requested by reviewer 2, point 4). In WT animals, we observe sustained H-events (**Fig. 2b,e**) comprising long duration calcium activity - reflected in the summary data shown in **Fig. 2e-g**. In KO animals, H events are more fragmented, exhibiting a reduction in both duration (**Fig. 2c,g**) and amplitude (**Fig. 2e,f**).
- The case for hypo-activity is further made by the unbiased assessment of total $\Delta F/F$ (**Fig.2j**), likely driven by the decrease in amplitude and duration of H-events. We have included discussion of the likely mechanisms underlying this hypoactivity incorporating previous literature later in the paper (page 26, line 585-603).

3. In figure 3, it is unclear why the authors did not probe potential changes in PV's inputs to PYR. This experiment will be essential for and strengthen the authors' argument since the conclusion is that SST's input to PYR is increasing to compensate for PV's decreased input, yet PV interneurons were never examined. How does elevated serotonin in SERT-KO pups impact the early development of PV?

This point is similar to one raised by reviewer 2, point 5

- Unfortunately, we cannot directly test the impact of elevated serotonin on PV interneuron signalling this as endogenous expression of the marker PV is delayed until the second postnatal week (e.g., Anastasiades et al., 2016) with the PV-Cre only giving reliable recombination at this time. Given this limitation, we set out to use a subtractive analysis approach using the *Nkx2-1-Cre* and *SST-Cre* data to infer the contribution of PV cells to the local circuit (*now* **Figure 4**) during the early postnatal Th-SERT hypoactive period (see ref 54 for an alternative approach). If one assumes this genetic strategy is valid then our data suggest that PV interneurons have little or no synaptic input onto pyramidal cells in SERT-KO pups (**Figure 4d,e**); an observation that is mirrored by the absence of local L4 GABAergic input in our LSPS data (Figure 4k-m) - input that is mediated predominantly by PV interneurons (Koebl et al., 2013).
- However, the reviewers' point is interesting and led us to explore the nature of the data obtained from the *Nkx2-1Cre* line. This was further sparked by recent *in vivo* experiments (see ref 75) which suggest that minimal photoactivation of *Nkx2-1 Cre;Ai32(ChR2)* neurons can be used to *selectively* optotag interneurons that contribute

to feed-forward inhibition, typically mediated by PV basket cells. As a first pass, we compared the LED power used to evoke minimal IPSC responses across our pyramidal cells sample from both *Nkx2-1Cre* and *SST-Cre* lines (**Extended Data Fig. 3e**), which revealed a significant difference in the power used for these 2 genetic tools. A finding that - assuming equal expression of ChR2 from the *Ai32* line - suggests that SST interneurons are selectively recruited at higher LED power. If so, we would expect the maximal response to reflect a combination of *Nkx2-1* and *SST* data, which is indeed the case (**Extended Data Fig. 3f**). We further performed *ex vivo* analysis (**Extended Data Fig. 3g-j**) wherein we targeted (ChR2)YFP-expressing interneurons in L2/3 based on their gross dendritic morphology under the patch clamp set-up: multipolar, non-adapting, putative PV cells or bitufted, adapting, putative *SST* interneurons (**Extended Data Fig. 3g**). We further confirmed their identity *post hoc* based on spike phase plot analysis (see ref. 57)(**Extended Data Fig. 3h**). Similar to our summary data from across all our optogenetic experiments, these two populations showed differing sensitivity to 470 nm LED stimulation with putative PV interneurons exhibiting larger inward current at low LED powers when compared to putative *SST*, bitufted interneurons (**Extended Data Fig. 3i,j**).

- The sum of these data suggest ChR2 expression in early postnatal *Nkx2-1* interneurons allows us to selectively record putative PV interneuron-mediated IPSCs in postsynaptic PYRs at minimal stimulation.

4. In figure 3, the authors should measure mIPSCs on L2/3 pyramidal neurons from each genotype (WT, Het, KO) to confirm that the changes in ChR2-evoked IPSCs are not due to alterations of inhibitory post-synapses on pyramidal neurons in Het or KO condition (2c, d and e). Inhibitory synapse development (post-synapse number and/or strength) could have been affected in SERT-KO mice.

- Our study used optogenetic and laser scanning photostimulation to delineate various changes to GABAergic interneuron connectivity over development. **Fig. 4** provides a comprehensive assessment of the circuit changes, pinpointing altered feed-forward inhibition - likely mediated by PV interneurons - as a key player in the *in vivo* phenotype described in the rest of the manuscript. At early postnatal ages, mIPSC frequency is so low that meaningful analysis is challenging. We cannot nor wish to exclude postsynaptic alterations but suggest further exploration of the precise locus of the serotonergic neuromodulation is beyond the scope of the current study, which is focused more on population level changes resulting from elevated serotonin. Indeed, serotonin is a pleiotropic modulator that acts on a wide variety of receptors as evident by the contrasting effects on putative PV and VIP interneurons reported in our manuscript.

5. In figure 4, does postnatal fluoxetine treatment increase serotonin levels in the cortex? If it does, is this increase diminished during the Th-SERT period, but not Adult SERT stage? This will be very important evidence supporting the buffering role of Th-SERT in cortex.

- The reviewer makes a good point and we thank them for raising this. We addressed this using the serotonin sensor g5-HT3.0 in mice dosed postnatally (P2-14) with sucrose or fluoxetine (**Extended Data Fig. 5a**). These experiments confirmed the reviewer's hypothesis, there is a significant increase in serotonin signal in SSRI dosed animals during the later, Adult-SERT time window. Whereas at early ages (Th-SERT) there is only a trend, likely reflecting higher SERT availability during this period. We have updated the text accordingly (page 18, line 407-412).

6. In figure 4i, sucrose treated mice showed substantial reduction in GCaMP6s signals, which could have led to a significant difference in GCaMP6s comparison between SSRI and sucrose treated groups. This reduction was not observed in figure 2j (I assume the same experimental conditions with the same ages). Control (sucrose) data must be well validated first to determine hypo vs hyper-excitability.

- The experimental conditions between **Fig. 2j** and **Fig. 4i** are not the same. The data shown in figure 2 labelled with a 'WT' genotype come from a cross between a pair of SERT heterozygous breeders. We report a difference in the underlying GABAergic circuitry due to maternal genotype (**Figure 5**) that could underlie this disparity. It is possible that the significant decrease observed after the initial sensory response in SSRI control (sucrose) animals is mediated by PV interneurons - interneurons that are variously compromised in the SERT WT and SSRI-dosed pups. The emergence of an inhibitory response at these ages (immediate Adult SERT) is consistent with increased perisomatic inhibition at this time reported both in sensory cortices (ref 51) and hippocampus (ref 66).

7. In figure 5l and m, the authors should show % SST, % PV, and % Nkx2-1 together.

- We have now quantified % VIP % SST, % PV, and % Nkx2-1 for Sucrose/SSRI treated mice, as well as % VIP % SST, and % PV in SERT-WT, Het, and KO mice. The % Nkx2-1 could not be quantified in SERT-KO mice as none of these animals survived past P10. All data has been added to the requested panels (l and m) of Figure 7 (old Figure 5).

8. The figures are very difficult to follow without simultaneously reading the text together with the figure legends. The authors sometimes add the postnatal age of the mice above the figure and sometimes it is just referenced in the text, or sometimes in the figure legends. For example, figure 2 e-h are from P7-10. Figure 2 j is from P11-13. Figure 2 m and o are from P7-10. Also, sometime Th-SERT is indicated on a figure panel and sometimes not. Please show Th-SERT vs Adult-SERT for each panel, when applicable. Young vs old and Th-SERT vs Adult-SERT are the key features of this study.

- We have added panels identifying Th-SERT and Adult SERT in all figures as requested.

9. The entire discussion needs to be expanded significantly. The authors should elaborate on what they think about how serotonin signaling is involved in pyramidal neuron activity and interneuron activity in the somatosensory cortex, and why their activity is so special during development. In addition, more about SERT polymorphisms should be added since Th-SERT vs Adult-SERT represents a large component of the rationale for the research work. Additionally, the authors should clarify the relevance of SERT-Het and SERT-KO relative to the polymorphisms seen in clinically relevant populations, especially since different responses are observed in Het vs KO in some datasets (e.g., figure 2n)

- We have reworked the discussion as suggested by the reviewer to now include sections on (1) the role of serotonin signalling in the developing brain (page 26, lines 586-604), and (2) the relevance and complexities of our SERT data versus current knowledge of polymorphism (page 26-27, lines 606-628).

Minor comments:

1. In figure 1 d and e, serotonin levels were measured near VIP (subregions surrounding VIP). Please accurately define the “proximal” in the text. Did the authors observe similar results if they examine the signals only from VIP-INS? Also, in figure 1 d and e, please clearly indicate ROIs of their measurement. The images are very vague.

- Quantification was done in areas with VIP interneurons. We have corrected the text to make this more clear by replacing “proximal” with “in areas with”. We have also drawn green squares in the representative images 1d and 1e, indicating which regions were considered to have VIP interneurons. Figure 1 caption has been updated accordingly.

2. In figure 1 g, h, and i, are g5-HT3.0 data from WT during Th-SERT period? Please label the genotype and experimental stage.

- We apologise for the confusion. We have updated the figure, as well as its caption, to clearly identify that the recordings are from WT animals during the Th-SERT period.

3. In figure 2 n and p, please show time-locked changes in GCaMP6s for whisker stim experimental data across development. At least, time-locked changes of VIP from Th-SERT vs Adult-SERT similar to data in figure j. Statistical comparisons are not clear on

figure 2 n and p. Please also show H-event duration from VIP (Figure 2 m) and Nkx2-1 (Figure 2 o) of each genotype.

- We have inserted a new figure (**Fig. 3**) to include the requested data.

4. In figure 3 k, n, and o, please clearly label if WT and HET indicate the maternal genotype or pups' genotype.

- This data now forms a separate figure, **Fig. 5**. We have also included appropriate labels as identified by the reviewer.

5. In figure 5 a-e, SSRI treatment period should be clearly shown in an experimental schema and/or timeline.

- This schema including a timeline has been added (**Fig. 7a**).

6. On line 206, please define SERTKO-HOM.

- We thank the reviewer for identifying this typo, it has been corrected in the text. Throughout the manuscript we use SERT-KO to refer to animals with a homozygous knock-out of SERT.

7. On line 220, it should be I, not i.

- The text has been updated accordingly.

Reviewer 2

Introduction: It would be useful to provide a couple of sentences and references explaining the developmental changes that the serotonergic system is undergoing according to age. For example, the authors report that P11 to P13 reflect adult-like SERT expression. Isn't this expression brain area dependent? Please provide the necessary explanations and literature.

- We agree with the reviewer and have made appropriate changes to the introduction. Given the complex changes to the serotonin system through development, we have referenced both anatomical and electrophysiological changes, as well as SERT expression dynamics, all of which are critical for contextualizing our study. This is now included in page 1 (lines 54-63) of the introduction.

Results:

1. Line 80: The serotonin sensor g5-HT3.0 was injected into the somatosensory cortex of which mice? Do the authors mean the VIP-Cre;Ai9? This information should be clarified from the beginning of the paragraph and the authors should explain already from the beginning why they chose this mouse line to test serotonin signaling to help with the flow of the story.

- We apologize for the lack of clarity and have included additional information in the text on page 2, line 88-91.

2. Line 93: How do the authors define localized signaling and what the authors consider as proximal to VIP interneurons? A few details should be provided here concerning the responses proximal to VIP.

- Quantification was done in areas containing VIP interneurons (identified by conditional expression of tdTomato). We have corrected the text to make this more clear by replacing "proximal" with "in areas with". We have also drawn green squares in the representative images 1d and 1e, indicating which regions were considered to have VIP interneurons. Figure 1 caption has been updated accordingly.

2. It would be nice to see a Time-series video showing serotonin fluctuations. For example, the authors could provide along with the Extended Data Fig. 1a the corresponding video.

- Changes in g5-HT3.0 fluorescence were small and slow due to the intrinsic properties of the sensor. While the quantification revealed fluctuations, these are hard to appreciate by direct observation of individual responses. We have now stated that we are showing average responses in the relevant result sections and included a time series video (Supplementary Movie 1) with an example of stimulus-triggered average of $\Delta F/F$ serotonin signal during whisker stimulation of SERT-KO mice (Th-SERT period). We

have also added a representative calcium movie (Supplementary Movie 2), in which fluctuations can be observed more clearly.

How many mice were tested? Please provide the number of animals for each graph.

- This has been added at the end of the Fig.1 legend.

3. The authors suggest that SERTKO mice transition from cortical hypoactivity to hyperexcitability. They show that during the early postnatal time period KO mice show a decrease in the amplitude and duration of H-events (Fig. 2e and f). However, the mean H-event frequency in KO mice is higher than WT (Extended Data Fig. 2b) at P7-10. This is a rather important information that should be moved to the main figure. How do the authors explain this increase in frequency and how this is aligned with hypoactivity? What is the mean frequency of H-event in VIP mice for WT, Het and KO? The mean frequencies should be plotted for all conditions for VIP and Nkx2-1 mice.

- H-event frequency has been brought to the main figures both for SERT-KO (**Fig. 2f**) and SSRI data (**Fig. 6f**). H-event frequency and duration has also been added for VIP/Nkx2-1 neurons in both SERT-KO (**Fig. 3**) and SSRI (**Extended data Fig. 5**).
- The increase in frequency is likely due to the impairment of H-events, characterized by reduced amplitude and duration. This results in more fragmented activity and a rise in the number of shorter, smaller H-events (see **Fig. 2c**). This has been added to the discussion (page 26 lines 586-589).
- Hypoactivity is clear from the left shift of the cumulative probability distribution of $\Delta F/F$ values and very likely driven by the decrease in amplitude and duration of H-events. A discussion of the potential mechanisms underlying this hypoactivity - based on previous reports - has now been included in the discussion (page 26, line 589-604).

Images in b and l should become bigger. The cells are not visible, especially for the Nkx2-1Cre.

- We have increased the size of these panels as requested, now in **Fig.3**.

4. It would be useful to see representative rasterplots for a given population of simultaneously recorded neurons to appreciate the presence of L- and H-events. These graphs should be added along with the heatmaps in Fig. 2c and d.

- We thank the reviewer for this useful suggestion; they greatly help understanding of serotonin-mediated changes in H-events dynamics. These graphs have been added to **Fig. 2** (WT and SERT-KO) and **Fig. 6** (Sucrose and SSRI).

5. The recordings of SST interneuron inputs in Fig. 3 show an increase in the conductance of IPSCs in SERT-KO L2/3 pyramidal neurons. The authors suggest that elevated serotonin levels in SERT-KO pups impair the early development of PV

interneurons and this impairment is so severe that the observed increase in SST interneuron input cannot compensate, leading to hyperexcitability in L2/3 of the S1BF around the onset of active sensation. Recordings of PV interneuron inputs are essential to corroborate this hypothesis in WT, Het and KO mice.

- This point is similar to that raised by the first reviewer (point 3) and is addressed in our response to them. In brief, PV expression is not present until the second postnatal week, precluding use of this marker to target early basket cells. We have however performed additional experiments and analysis to validate use of the Nkx2-1 line to target these cells during postnatal life.

6. The VIP amplitude during H-event after SSRI dosing is not altered (Fig. 4 and Extended Fig. 4). How do the authors explain this difference in comparison to the strong effect they found for the KO mice?

The reviewer highlights an important difference between the observations in SERT-KO and SSRI-treated mice. We have now expanded the H-event analysis in both VIP and Nkx2-1 interneurons for both datasets, including duration and frequency of H-events in each subpopulation (**Fig. 3** and *now Extended Data Fig. 5*). While there are similarities in the results between the genetic and pharmacological models (e.g., the switch from hypo- to hyper-activity) we also observed several differences such as the increased amplitude of VIP interneuron recruitment in SERT-KO but not SSRI-treated pups as highlighted by the reviewer. One plausible explanation is that highly synchronous H-events are more impacted in the latter, attenuating local drive onto these interneurons. These inconsistencies observed between SERT-KO and SSRI effects in postnatal development likely stem from the temporal difference of these SERT disruptions (postnatal versus life-long) and the nature of the disruption (genetic versus pharmacological).

We have added text to the relevant results and discussion sections to highlight these complexities (page 19, line 437-440 and page 26-27, line 619-625 etc.).

The number of recorded neurons should be indicated along with the number of mice.

We thank the reviewer for pointing out this omission. We have now included these details in the figure caption for **Extended Data Fig. 5**.

7. The authors show that postnatal SSRI administration results in long-lasting effects on sensory encoding. They show that single-whisker stimulation or airpuff presentation increased cortical response amplitude and neuronal recruitment in SSRI treated animals but not in the KO and they hypothesize that this is due to a prolonged impairment of the inhibitory control of information transfer. They present stainings for VIP and PV interneurons. SST quantifications should be added in a similar manner for WT, Het and KO mice since these neurons have a critical role in cortical circuitry as it was also suggested in Fig. 3 to support their results.

- Immunohistochemistry for the neuropeptide SST is now included for both SERT-WT/Het/KO and Sucrose/SSRI postnatally treated mice (Figure 7). No significant differences are observed as previously reported by reference 68.

8. It will be nice to see analysis of synchronous events in the adult mice of 8-12 weeks that are presented in Fig. 5 to evaluate possible alterations that may occur postnatally in later ages as a result of SSRI exposure and maternal SERT disruption. This figure could be added to the supplementary material of the paper for the different conditions of single whisker deflection and airpuff stimulation.

- H- and L-events are characteristic synchronous calcium events observed in the developing neocortex, but absent post-P10, following the transition to desynchronous activity at P11 (ref.13). However, as highlighted by the reviewer, changes in the synchronicity of cortical activity might be present in the adult cortex of these mice. We have computed the Pearson's correlation coefficient and constructed correlograms of all cells. Representative correlograms for each group have been included, as well as average data, for both single whisker deflection and airpuff stimulation (**Extended Data Fig. 7**). The data shows a significant increase in neuronal correlations during adult airpuff stimulation in SSRI postnatally-treated mice. The text has been updated to reflect this (page 20 line 471-473).

9. Did the authors see any differences on serotonin signaling among male and female mice? It would be very interesting to see if the dynamics change in relation to the sex of the mouse since they authors used both sexes for their study. If this is not possible, at least it should be discussed in the discussion paragraph.

- The reviewer raises an interesting point given the previous literature addressing differences in the adult serotonergic system of male and female mammals. Unfortunately, our study lacks sufficient power to address the reviewer point if the data is split by animal sex, we included a supplemental figure showing our main results separated by male *versus* female data (**Extended data Fig. 11**), for both serotonin and calcium dynamics. Overall the same trends were observable with minor differences, suggesting that our results are common to both males and females. We now refer to this figure in the discussion (page 27 line 625-628), as we hope it will serve to inspire future studies addressing sex differences in the dynamics and effects of serotonin in the developing cortex.

Other comments:

10. Why did the authors image tdTom at 765nm (line 578)? With an absorption peak around 554nm and emission peak around 581, one would expect that for 2P imaging the authors would image tdTom above 1000 nm.

- The reviewer brings up a relevant concern since indeed most two-photon imaging studies employ wavelengths of $>1000\text{nm}$ for red-shifted fluorophores, as sensors generally show a 2p excitation peak and their highest dynamics in this range. However, most red sensors (including tdTomato) present a second excitation peak at 700 nm (see <https://www.fpbases.org/protein/tdtomato/>). This wavelength is generally not used because the dynamics (i.e., their differential emission bounded or unbounded to their target molecule, such as calcium) of most red-shifted sensors (e.g., jRCaMP1a) are relatively poor at this imaging wavelength. In our case, we were only imaging a fluorophore (tdTomato) without biosensor capabilities, which is why this wavelength was a valid approach. Given its validity, we chose this wavelength for practical reasons, which is that we image with a tunable Chameleon Ultra II Laser that cannot be tuned above 1000 nm.

Line 603: I would suggest to replace Fmean by F0 to avoid confusions. Otherwise, the authors should clarify at line 604 that Fmean is the mean baseline fluorescence of the recording session.

- While we acknowledge the source of confusion, we maintained Fmean because some of the panels represent analysis of baselines (e.g., **Fig. 1k**), in which case the normalization reflects the whole recording and not a prior time as F0 would indicate. We thank the reviewer for detecting this and we have updated the text accordingly (page 31 line 756).

11. How did the authors account for movement in the z-plane for the analysis of the two-photon calcium imaging data? A description should be added to the Methods part.

- We have now added this information to methods (page 30 line 730-733), and further included a video (Supplementary Movie 2) showing an unregistered calcium recording during animal movement to illustrate recording stability.

12. For all figures, please provide the number of cells and number of mice recorded and analyzed.

- This change has been implemented across all figures.

RESPONSE to REVIEWERS

Reviewer #2:

The authors have properly addressed most of my concerns by adding new experiments and editing the text. It is a beautiful work.

We thank the reviewer for their continued support of our work.

However, I still have doubts if the authors can claim on their ms that the SERT “transiently” expresses during early development “without” showing the “transient” expression of SERT in their system. The references they cited are good and sufficient for making a rationale for their work, but not for demonstrating that the SERT is indeed expressed transiently in S1BF.

Fig S5 A might indirectly support it though. What is the age of slice on the panel B of Fig S1? I would show the similar images of SERT expression from Th-SERT (I assume the current Fig S1 B) and Adult-SERT, side by side, in order to claim that SERT expresses “transiently” in S1BF.

It had been shown by a number of groups, notably Gaspar (e.g. Lebrand et al., 1996, 1998) and Hoffman (e.g., Hansson et al., 1998) that both SERT mRNA and protein are transiently expressed in rodent postnatal telencephalon, notably primary sensory thalamic nuclei, with mRNA peaking at postnatal day (P)7 and fading by P10. We recommend the Lebrand et al., 1998 paper (J Comp Neurol 401(4):506-24) as it provides a detailed assessment of SERT expression in the rodent CNS across development.

The image shown in Extended Data Fig. 1b was generated using the SERT-Cre line which is highly effective in *fate-mapping* all cells that express SERT through life (see Narboux-Nême et al., 2008).

Given the reviewer’s concerns, we have further performed immunohistochemistry for SERT (similar to Lebrand et al., 1998) and included this as a new panel Extended Data Fig. 1c in our manuscript. Expression of SERT protein is clearly seen in the barrels at P7 in wild-type (WT) pups but fades by P14. No protein was detected at any age in the SERT-KO littermate.

Finally, the reviewer might be interested to know that transient expression of SERT protein is routinely used by developmental neuroscience labs as an immunohistochemical marker for thalamocortical afferent fibres in the early postnatal brain (e.g., Hoerder-Suabedissen et al., 2008; Chen et al., 2016; Mercurio et al., 2019).

Minor points

1. * on VIP of Fig1 D might be an error. Its legend indicates $p = 0.007$ (line 155).

We thank the reviewer for spotting this and have corrected panel 1D

Line 173: * $P < 0.05$, ** $P < 0.05$ seems to be a typo.

A good spot! We have corrected this to ** $P < 0.01$.

2. Figure 1 B and C, airpuff (black) and whisker (dark gray) are clearly indicated, but the data point/line for sound (light gray) is not clearly shown.

We agree that the grey was less than optimal and have changed to orange to avoid a colour clash with other data.

3. Figure 2, ANOVA (lines 216, 223, 226) needs to be clarified. One way or Two way.

The Anova were two way and these clarifications have now been added to the legend in the appropriate place.

4. Figure 3 A, labeling (or color-coded) bottom-up, feed-forward, and top-down inhibition will be helpful for readers.

We are happy to do this and have amend this diagram to show bottom-up in red and top-down in green

5. Line 342: not h but j.

Another good spot and we have corrected this.

6. Figure 4 M, showing HET data will be useful for comparison.

We have not shown and would prefer to continue to not show the SERT-Het profile to ease comparison of the WT and KO profiles. The Het data is a broader version of the WT with a peak in L4, extending into L5 as can be deduced from the average map shown in the middle panel of 4L.

7. The authors could add “in WT” on Figure 6 title. For example, Postnatal fluoxetine treatment in WT recreates the SERT-KO transition from cortical hypoactivity to hyperexcitability.

We thank the reviewer for this suggestion. It is an important clarification and will assist the reader.

Reviewer #3:

The authors have significantly revised the manuscript by performing additional analyses and incorporating more data into the main figures, including additional Extended Data Figures. Furthermore, they have substantially enhanced the discussion section. I have no additional comments.

We thank the reviewer for taking the time to review and improve our manuscript.